# VLM Agents Generate Their Own Memories: Distilling Experience into Embodied Programs of Thought

**Gabriel Sarch**[1]   **Lawrence Jang**[1]   **Michael J. Tarr**[1]

**William W. Cohen**[1,2]   **Kenneth Marino**[2]   **Katerina Fragkiadaki**[1]

[1]Carnegie Mellon University   [2]Google DeepMind

https://ical-learning.github.io

## Abstract

Large-scale generative language and vision-language models (LLMs and VLMs) excel in few-shot in-context learning for decision making and instruction following. However, they require high-quality exemplar demonstrations to be included in their context window. In this work, we ask: Can LLMs and VLMs generate their own examples from generic, sub-optimal demonstrations? We propose In-Context Abstraction Learning (ICAL), a method that builds a memory of multimodal experience from sub-optimal demonstrations and human feedback. Given a task demonstration that may contain inefficiencies or mistakes, a VLM abstracts the trajectory into a generalized program by correcting inefficient actions and annotating cognitive abstractions: causal relationships, object state changes, temporal subgoals, and task-relevant visual elements. These abstractions are iteratively improved and adapted through human feedback while the agent attempts to execute the trajectory in a similar environment. The resulting examples, when used as exemplars in the prompt, significantly improve decision-making in retrieval-augmented LLM and VLM agents. Moreover, as the agent's library of examples grows, it becomes more efficient, relying less on human feedback and requiring fewer environment interactions per demonstration. Our ICAL agent surpasses the state-of-the-art in dialogue-based instruction following in TEACh, multimodal web agents in VisualWebArena, and action anticipation in Ego4D. In TEACh, we achieve a 12.6% improvement in goal-condition success. In VisualWebArena, our task success rate improves over the SOTA from 14.3% to 22.7% using GPT4V. In Ego4D action forecasting, we improve over few-shot GPT-4V and remain competitive with supervised models. We show finetuning our retrieval-augmented in-context agent yields additional improvements. Our approach significantly reduces reliance on manual prompt engineering and consistently outperforms in-context learning from action plans that lack such abstractions.

## 1   Introduction

Humans exhibit remarkable few-shot learning capabilities, rapidly generalizing from a single task demonstration to related conditions by integrating the observed behavior with their internal world model. They discern what is relevant and irrelevant for success and anticipate potential failures. Through repeated practice and feedback, they quickly find the right abstraction that helps to imitate and adapt the task to various situations. This process facilitates continuous refinement and transfer of knowledge across a diverse range of tasks and contexts.

38th Conference on Neural Information Processing Systems (NeurIPS 2024).

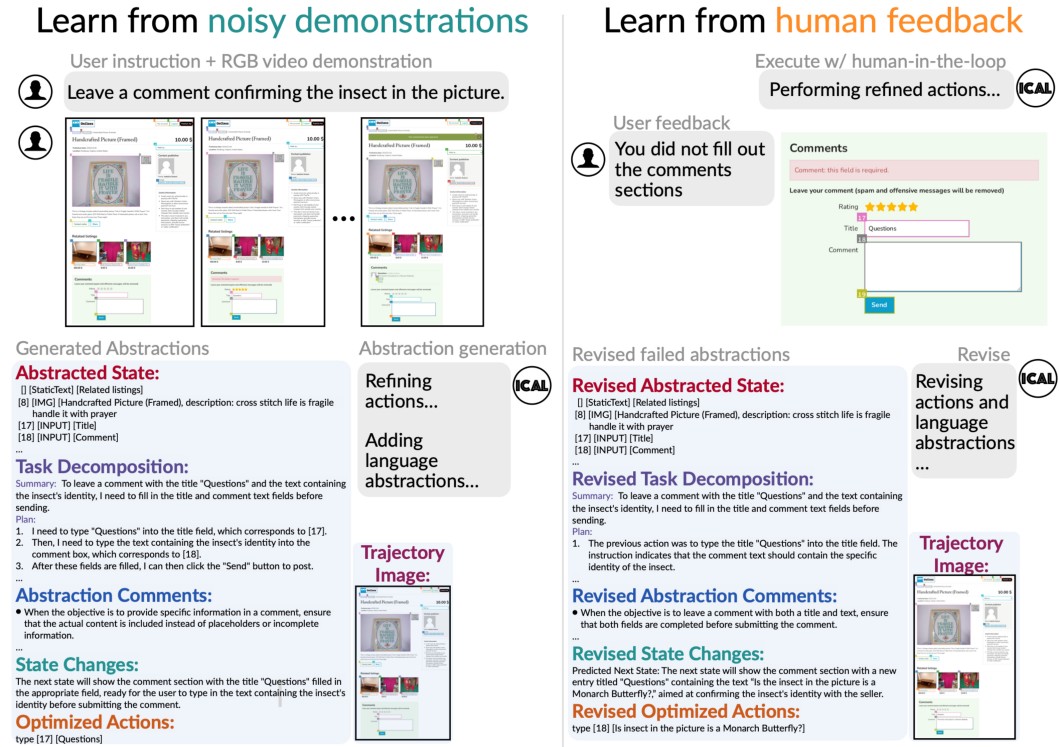

Figure 1: ICAL (In-Context Abstraction Learning) is a method for efficient agent learning from both noisy visual demonstrations and human feedback using large language / vision models. *Left:* The agent can take in a video demonstration, and generate a refined example with language annotations to be used later by the VLM via in-context learning. *Right:* Humans provide feedback, correct errors and supply additional knowledge.

Recent research has explored the use of large language models (LLMs) and visual-language models (VLMs) [1] to extract high-level insights from trajectories and experiences. These insights are generated through the model's introspection and are used to enhance performance by appending them to prompts, leveraging their strong in-context learning abilities [39, 70, 56, 60]. Existing methods often linguistically focus on task reward signals [70, 56, 76, 79], store human corrections following failures [88, 15, 68], use domain experts to hand-write or hand-pick examples without introspection [68, 73], or utilize language to shape policies [30, 74] and rewards [61, 3, 27, 21, 26, 59, 74, 35, 54]. Critically, these methods typically are text-based and do not incorporate any visuals cues or demonstrations, or use introspection only in case of failures, which is only one of several ways that humans and machines can consolidate experiences and extract insights.

**In this work, we teach VLMs novel tasks by learning in-context experience abstractions given sub-optimal demonstrations and human natural language feedback**. We present In-Context Abstraction Learning (ICAL), a method that prompts VLMs to create multimodal abstractions for unfamiliar domains. Unlike previous works that only store and retrieve successful action plans or trajectories [68, 76, 44], our approach emphasizes learning abstractions that encapsulate the dynamics and critical knowledge of tasks, as illustrated in Figure 1. Specifically, ICAL tackles four types of cognitive abstractions: **task and causal relationships**, which identify the fundamental principles or actions needed to achieve a goal and how elements are interconnected through cause and effect [75]; **changes in object states**, which describe the various forms or conditions an object will take [4]; **temporal abstractions**, which break down tasks into subgoals [6]; and **task construals**, which highlight critical visual details within a task [31]. When provided with optimal or suboptimal demonstrations, ICAL prompts a VLM to transform these demonstrations into optimized trajectories

---

[1]Throughout the remainder of the paper, we refer to multimodal large language models capable of processing both text and images (e.g., GPT-4V) as 'VLMs'.

while also creating pertinent language and visual abstractions. These abstractions are then refined through executing the trajectory in the environment, guided by natural language feedback from humans. Each step of abstraction generation leverages previously derived abstractions, enabling the model to improve not only its execution but its abstraction capabilities as well. Collectively, the learned abstractions summarize crucial information about action sequences, state transitions, rules, and focus areas, articulated through free-form natural language and visual representations.

We present a comprehensive evaluation of our agent, equipped with the learned example abstractions, across three benchmarks: TEACh [63] for dialogue-based instruction in household settings, VisualWebArena [37] for multimodal autonomous web tasks, and Ego4D for video action anticipation [28]. In TEACh, our agent sets a new state-of-the-art, outperforming VLM agents reliant on raw demonstrations or extensive domain-expert hand-written examples, demonstrating the effectiveness of ICAL learned abstractions for in-context learning. Specifically, our approach achieves a 12.6% improvement in goal condition success compared to the previous SOTA, HELPER [68]. We show that this approach leads to increasing performance gains on unseen tasks as the external memory grows, and achieves a 14.7% performance increase after only ten examples. Moreover, our agent becomes increasingly efficient over time by leveraging stored abstractions, requiring 38.8% fewer environment steps and 71.6% less human feedback per example in the latter half of demonstrations processed. Integrating our learned examples with LoRA-based fine-tuning of an LLM [32] further improves goal-condition performance by 4.9%. In the VisualWebArena, our agent surpasses the state-of-the-art, GPT4 + Set of Marks [37], improving from 14.3% to 22.7% using GPT4V and from 18.9% to 23.4% using GPT4o. In the Ego4D setting, ICAL outperforms few-shot GPT4V using chain of thought, reducing the noun and action edit distance by 6.4 and 1.7, respectively, and competes closely with fully supervised methods, despite using 639x less in-domain training data. Our approach significantly reduces reliance on expertly-crafted examples and consistently outperforms in-context learning from action plans or trajectories that lack such abstractions [68, 76, 44].

## 2   Related Work

**VLM Agents**    LLMs and VLMs trained from large scale vision-language data have been adapted for task planning and decision making tasks through in-context prompt optimization or finetuning. VLMs have been used to plan over high-level actions or code [80, 76, 68, 44, 72], incorporate error feedback [52, 45, 88], and understanding game instruction manuals [83]. Some studies use VLMs for learning from human feedback through retrievable knowledge [88], question asking [66, 15], or converting language to actions or rewards [49, 50, 36, 11, 14]. Our work utilizes noisy visual demonstrations, and integrates multiple types of multi-modal abstractions during the learning process.

**Instructable Interactive Agents**    Benchmarks for embodied instruction following include question answering [25, 16, 93, 18, 17, 23], navigation [42, 41, 10], interactive dialogue, and instruction following [86, 71, 63, 22]. Virtual agent benchmarks focus on web tasks where agents navigate static [53, 19] and dynamic web environments [92, 37, 85, 38], covering personal shopping, travel assistance, software engineering, and operating system tasks [51, 34, 69, 47]. This includes visual grounding and multi-turn planning, with prior studies using finetuning or few-shot prompts. In agent-based domains, retrieval-augmented prompting and prompt optimization have improved task planning in instructional contexts [73] and open-world gaming [79, 76, 56, 62]. Unlike studies that rely solely on static external memory or text-based prompting, our research demonstrates that multi-modal, generalizable abstractions learned from a few noisy trajectories and human feedback via in-context learning or finetuning can significantly enhance instruction-following performance.

## 3   In-Context Abstraction Learning (ICAL)

In-Context Abstraction Learning (ICAL) aims at automating the acquisition of generalizable examples and knowledge for in-context agents. ICAL operates by receiving a language instruction $I$ with a noisy trajectory of observations and actions, denoted $\xi_{noisy} = \{o_0, a_0, \ldots, o_T, a_T\}$ in a new task domain $D$. A new domain $D$ represents changes in task variables not captured in VLM pretraining, such as a different environment (e.g., kitchen #1 vs. kitchen #2), task (e.g., "add the cheapest red bike to my wish list"), or user preference (e.g., "I prefer the red cup for coffee"). The core aim of ICAL is to abstract each noisy trajectory into a single example $e$, which then forms part of a memory set

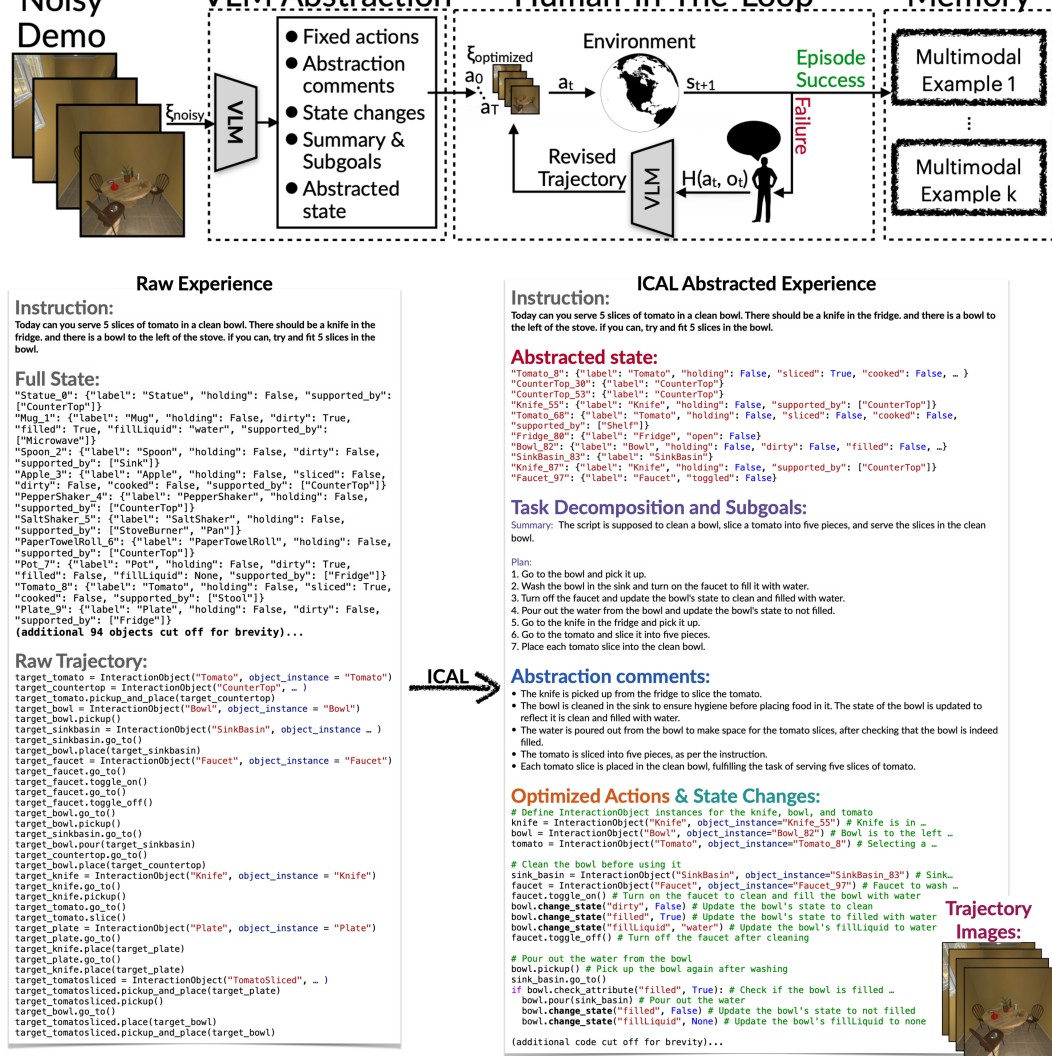

Figure 2: ICAL transforms raw experience into useful abstractions for in-context learning. *Top:* Given a noisy trajectory, It prompts a VLM to optimize actions and add language annotations. The optimized trajectory is executed, incorporating human feedback on failures. Successful examples are stored for future VLM in-context action generation. *Bottom:* An example of the raw, noisy trajectory (left), and the final abstracted example after ICAL (right).

M. Each example $e \in M$ represents an optimized trajectory $\xi_{optimized}$ with generalizable language abstractions $L$. The objective is to ensure that M collectively encapsulates examples that, when used in a VLMs context window, increase the likelihood of successful task execution in the new domain, while also containing knowledge that is transferable across similar tasks and contexts. This can be encapsulated as:

$$\max_{M} \mathbb{E}[R|M, I, o_t, D], \tag{1}$$

where $R$ is the return or the cumulative reward acquired by performing actions based on the instruction $I$, observation $o_t$, and in-context example memory set M. Rather than using reinforcement learning to optimize prompt examples through trial and error—which would lead to a challenging search problem that myopically focuses on improving rewards for the current scene—we leverage VLMs' knowledge for abstraction, which we elicit through prompting.

## 3.1 Overview

Figure 2 shows an overview of ICAL. Each iteration starts with a noisy trajectory. ICAL abstracts it in two phases: (1) abstraction phase ($F_{abstract}$), where a VLM corrects errors and enriches the sequence with language comments (Section 3.2). During this phase, a VLM identifies and corrects errors within the sequence, as well as enriches it with natural language comments. (2) The human-in-the-loop phase, denoted $F_{hitl}$, during which the sequence is executed within the environment and its abstraction is guided by human feedback conveyed in natural language (Section 3.3). Upon the successful execution of the trajectory, it is archived within a continually growing repository of examples. These examples serve as contextual references for the agent both during its learning phase and during inference for unseen instructions and environments.

## 3.2 VLM-driven Abstraction Generation

We address the challenge of learning from a diverse set of noisy trajectories $\xi_{noisy} = \{o_0, a_0, \ldots, o_T, a_T\}$, which may be sub-optimal due to several factors: demonstrations by human non-experts, errors in inferring actions from visual passive demonstrations, and generated paths that include exploration or failures. Please see Section 4.1 for details on noisy trajectory collection.

Abstracting a noisy trajectory, $\xi_{noisy}$, involves transforming it into a more optimized sequence, $\xi_{optimized}$, and formulating relevant language abstractions, $L$, as shown in Figure 2. The abstraction function, $F_{abstract}$, modifies $\xi_{noisy}$ by correcting actions and generating language abstractions that encapsulate general knowledge and task-specific insights. It is defined as:

$$F_{abstract} : (\xi_{noisy}, I, \{e^1, \ldots, e^k\}) \rightarrow (\xi_{optimized}, L) \tag{2}$$

where $\xi_{noisy}$ is the initial noisy trajectory, $I$ is the task instruction, and $\{e^1, \ldots, e^k\}$ are the top-k previous successful in-context examples. The output consists of the optimized trajectory $\xi_{optimized}$ and language abstractions $L$.

Corrections during abstraction include action adjustments and generating annotations ($L$) for abstracting subgoals, causal relationships, state changes, and reasoning steps. These annotations are produced by prompting the VLM to output a specified type of abstraction. We prompt the VLM abstraction function, $F_{abstract}$ (GPT4V in this work), to produce the abstractions detailed below. For the complete prompts, please refer to the Appendix.

**1. Task and Causal Abstractions:** Task and causal abstractions pinpoint the essential principles or actions required to achieve a goal and explain how elements are interconnected through cause and effect. Task and causal abstractions have been shown to be helpful in improving LLM generalization [56], and play a strong role in human communication and learning [75, 24]. We prompt the VLM to add annotations of task and causal abstractions in the form of natural language comments. For example, it might add a note explaining unnecessary actions, such as "Since the box is already open, there is no need to close it after placing the watches inside, ensuring the task is completed efficiently."

**2. State Changes:** Understanding how one's actions will affect the form and conditions of elements in a scene is crucial for decision-making [4]. The VLM is prompted to identify and predict state changes that occur during the demonstration. For instance, an annotation might note the bowl becoming clean, clearly indicating an expected state transition.

**3. Task Decomposition and Subgoals:** Breaking down a complex task into intermediate steps and subgoals is crucial for managing extended and variable sequences of lower-level actions. These temporal abstractions are important for human reasoning [6] and have been shown to improve LLM outputs [81]. We prompt the VLM to add 1) a step-by-step plan detailing the demonstration, and 2) a natural language summary of the actions.

**4. State Abstraction:** Useful representations do not simply mirror every aspect of the world; instead, they selectively capture a manageable subset of details relevant to a specific purpose [31]. We focus on identifying and including only those state variables that are relevant to the task at hand. This is achieved by (1) selecting parts of the state that were directly interacted with by the agent during the demonstration, and (2) prompting the VLM to suggest additional state variables not explicitly included in the demonstrations but potentially relevant to understanding the task.

### 3.3 Abstraction Verification with a Human-in-the-loop

In this phase, ICAL verifies the generated abstractions with a human-in-the-loop. This involves executing the optimized trajectory, $\xi_{optimized}$ within the actual task environment, under the watchful guidance of a live human observer. The procedure is:

**1. Execution of optimized trajectory:** The agent attempts to perform the task by following the optimized sequence of actions $\xi_{optimized}$ from the abstraction phase.

**2. Monitoring and Intervention:** As the agent executes $\xi_{optimized}$, a human observer monitors the process. If an action $a_t$ fails, denoted by $F(a_t) = 1$, the observer intervenes by providing natural language feedback $H(a_t, o_t)$. This feedback is context-specific, addressing the observed failure directly (e.g., explaining that the Toaster is currently full and can only toast one slice of bread). We provide additional details on the human-in-the-loop in the Appendix Section S5.1.3.

**3. Feedback Integration and Trajectory Revision:** Upon receiving feedback $H(a_t, o_t)$, the VLM is provided with this input alongside the current state of $\xi_{optimized}$ and any existing language annotations $L$. The VLM is prompted to revise $\xi_{optimized}$ to address the failure, to update existing annotations $L$ based on the feedback, and to add new annotations that capture insights from the feedback.

This process can be represented by an update function:

$$\Xi_{update}(\xi_{optimized}, H(a_t, o_t), L, I, \{e^1, ..., e^k\}) \rightarrow \xi'_{optimized}, L' \tag{3}$$

where $\Xi_{update}$ denotes the update function that takes the current trajectory $\xi_{optimized}$, human feedback $H(a_t, o_t)$, and current annotations $L$, and outputs the revised trajectory $\xi'_{optimized}$ and updated annotations $L'$. For the complete prompts, please refer to the Appendix.

**4. Environment Reset and Retrial:** Following a failure and subsequent feedback, the environment is reset to a suitable state for retrying the task. The agent then attempts the task again, utilizing the newly revised trajectory $\xi'_{optimized}$.

**5. Success Criteria and Feedback Limit:** This interactive phase continues until the human observer deems the task execution successful, or until a predefined maximum number of feedback iterations, $N_{feedbacks}$, has been reached.

**6. Saving example:** If successful, we store the revised trajectory $\xi_{optimized}$ and language annotations $L$ to the memory set M. If unsuccessful after $N_{feedbacks}$ iterations, we do not store the example and move to the next demonstration. We experiment with relabeling partially successful demonstrations in Section S4.5 of the appendix.

### 3.4 Retrieval Augmented Generation at Deployment

Given the learned example set M and a new instruction $I$, we prompt the VLM to carry out the instruction by producing action sequences $\{a_0, ..., a_T\} \in A$ from an action API that describes the skills set $A$ (e.g., `go_to(X)`, `pickup(X)`), by retrieving the top $K$ examples from M to include in the prompt based on their textual and visual similarity with the current scene. The aggregated similarity score $s$ for each example $e$ reads:

$$s = \lambda_I \cdot s^I + \lambda_{\text{textual}} \cdot s^{\text{textual}} + \lambda_{\text{visual}} \cdot s^{\text{visual}}, \tag{4}$$

where $s^I$, $s^{\text{textual}}$, and $s^{\text{visual}}$ are the similarity scores for the input text instruction, textual state, and visual state, respectively, computed via cosine similarity using embeddings from OpenAI's text-embedding-ada-002 model and CLIP ViT-B/32 model. The coefficients $\lambda_I$, $\lambda_{\text{textual}}$, and $\lambda_{\text{visual}}$ are weighting hyperparameters chosen in each domain by a held out validation set.

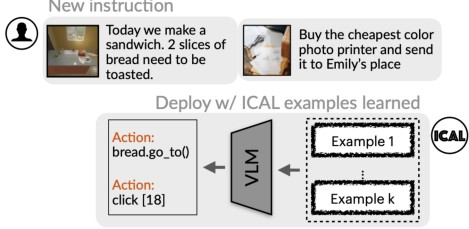

Figure 3: After the ICAL examples have been learned, ICAL is deployed for new tasks and environments using retrieval-augmented generation.

The VLM prompt contains the new instruction $I$, the current webpage image for web agents or 12 video frames for ego4D annotated with set-of-marks [84], a textual state description $x_t$ describing

the objects and their attributes for embodied agents and HTML elements for web agents, the action API $A$, and the retrieved set of in-context examples $e^1, ..., e^k \in M$. An illustration of this process is shown in Figure 3. The deployment prompt is provided in the Appendix.

**Implementation details** We use `GPT-4-1106-preview` [1] for text generation, unless otherwise stated, and `text-embedding-ada-002` [29] for text embeddings. We use `gpt-4-1106-vision-preview` [1] for the text and image generation model. We use $k = 5$ for example retrieval. We use a temperature of 0 for TEACh and Ego4D, and 0.2 for VisualWebArena.

## 4 Experiments

We test ICAL for task planning in TEACh [63] and VisualWebArena [37] and for action forecasting in Ego4D [28] benchmarks.

### 4.1 Environments

**TEACh [63]** The TEACh dataset comprises over 3,000 dialogue-based instructions for household tasks in AI2-THOR [40]. We use the Trajectory from Dialogue (TfD) tasks where agents convert dialogue instructions into action sequences, such as MAKE COFFEE. It includes training and validation splits (seen and unseen), the latter featuring new environments and instructions. Agents receive egocentric image inputs $o_t$ and perform actions like `pickup(X)` and `turn_left()`. Task success is contingent on fulfilling all instruction conditions. Utilizing HELPER's [68] perception, navigation, and manipulation modules, the system relies on RGB images, depth maps, object masks, and egomotion for 3D mapping and object recognition. We remove domain-specific checks from HELPER's modules to allow ICAL to learn them independently. **Noisy Trajectories.** We use 250 noisy trajectories from TEACh, omitting action labels but retaining language instructions and corresponding RGB videos. To label actions from RGB video, we trained an inverse dynamics model using a transformer encoder-decoder based on the DETR architecture [7] from a seperate 300 TEACh episodes. Model predictions and human errors, like unnecessary movements, cause action noise in these demonstrations. 122 examples were successfully abstracted by ICAL.

**VisualWebArena [37]** VisualWebArena consists of 910 episodes across various web tasks (Classifieds, Shopping, Reddit) requiring visual comprehension and reasoning. Instructions may include text and reference images, like adding an item seen in an image to a wish list. Agents operate with instructions $I$, current webpage images, and an API for actions like `click(X)`, executing tasks to fulfill instruction conditions. **Noisy Trajectories.** From VisualWebArena, 30 human demonstrations and 62 model trajectories from few-shot GPT4V were abstracted using ICAL. The process led to an example set of 92 for evaluation.

**Ego4D [28]** This task involves anticipating actions from Ego4D RGB egocentric videos in daily scenarios. Models select from 115 verbs and 478 nouns for predicting actions. We evaluate using 200 unseen videos from ego4D validation, applying edit distance as a performance metric. Input to models includes sequences of video frames annotated with set-of-marks [84] tracking [12] and label masks. The supervised baseline [28] (243 video hrs of Ego4D V2) uses a SlowFast backbone with a Transformer aggregator. **Noisy Trajectories.** Due to the passive nature of this task, ICAL proceeds without human-in-the-loop verification during ICAL (only Section 3.2, VLM-driven Abstraction Generation). ICAL successfully abstracted 92/100 demonstrations taken from the Ego4D validation set (8 failed due to GPT filters) for evaluation.

### 4.2 ICAL beats written & unchanged demonstrations in household instruction following

Table 1 presents our findings on the TEACh unseen validation set, assessing performance on new instructions, houses, and objects. ICAL and baselines use HELPER's navigation and manipulation modules [68]. We compare with these baselines: *1. Hand-written examples* from HELPER, the SOTA on the TEACh benchmark, with 19 expert-written examples for retrieval-augmented prompting. *2. Zero-shot chain of thought*, prompting the LLM to output step-by-step. *3. Raw Visual Demos*, retrieving unchanged demonstrations labeled with the inverse dynamics model. *4. Raw Kinesthetic Demos*, retrieving unchanged demonstrations with true actions. Our metrics are: *1. Task success rate (SR)*, the % of tasks completed successfully. *2. Goal condition success rate (GC)*, the % partial fulfillment rate across sessions.

As shown in Figure 4, ICAL revises noisy trajectories, enabling more successful tasks completed on training tasks than mimicking raw trajectories, with increases of 42 and 86 successful tasks for kinesthetic and visual demonstrations, respectively. This shows how ICAL not only adds useful abstractions but also corrects errors in the passive video demos, improving success in the original demo environment. Please see the Appendix Section S4.3 for additional analysis.

As shown in Table 1, on unseen tasks, ICAL outperforms unprocessed demonstrations as in-context examples, achieving a 17.9% absolute improvement in SR over raw demos with predicted actions and 8.6% over those annotated with true actions. This underscores the effectiveness of our abstractions in improving the quality of examples for improved in-context learning, unlike previous works that primarily save and retrieve successful action plans or trajectories without abstractions [68, 76, 44].

Additionally, we surpass the handwritten examples of the previous SOTA HELPER [68] by 12.6% in GC and 0.6% in SR, and by 2.2% (relative 26.5%) using estimated perception , demonstrating our method's efficacy with less expert intervention, leveraging only visual demos and non-expert feedback. Unlike HELPER, which requires domain experts to write 48-107 lines of text for each example, ICAL does not rely on such extensive input from experts. Instead, it allows non-experts to provide up to five natural language feedback corrections to the agent, significantly reducing the required effort and expertise per example.

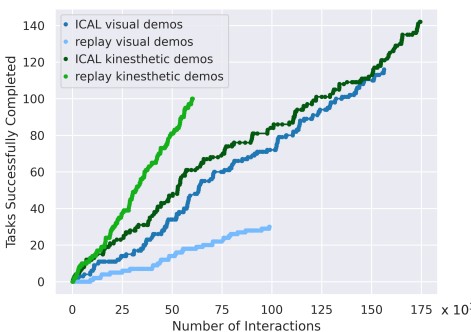

Figure 4: **ICAL enables greater success on training tasks.** Tasks successfully completed by ICAL over number of interactions when using the ICAL method with kinesthetic or visual demonstrations, and when replaying the kinesthetic or visual demonstrations directly.

Table 1: **Evaluation on TEACh unseen validation set.** All evaluations are done using GPT3.5-turbo-1106 unless otherwise noted. Visual Demos = demonstrations labeled with inverse dynamics model. Kinesthetic Demos = demos labeled with GT actions. GC = goal-condition success

|  | Success | GC |
| --- | --- | --- |
| *Ground truth segm, depth, attributes* | | |
| HELPER hand-written [68] | 34.5 | 36.7 |
| Zero-Shot CoT [39] | 11.8 | 24.6 |
| Raw Visual Demos | 17.2 | 26.6 |
| Raw Kinesthetic Demos | 26.5 | 29.5 |
| ICAL (ours) | **35.1** | **49.3** |
| w/o abstraction phase | 29.4 | 44.9 |
| w/o human-in-the-loop | 29.9 | 41.0 |
| w/ retrieval re-ranking | **35.3** | **51.7** |
| w/ GPT4 | 41.7 | 63.6 |
| | | |
| finetuned | 23.2 | 40.3 |
| finetuned + retrieval | **35.8** | **54.2** |
| *Estimated perception* | | |
| HELPER hand-written [68] | 8.3 | 14.1 |
| ICAL (ours) | **10.5** | **15.4** |

Table 2: **Results in VisualWebArena.** ICAL outperforms the prior best, GPT4o/V + Set of Marks. Ablation studies were conducted with GPT4V on a subset of 257 episodes.

|  | Seen | Unseen | Average |
| --- | --- | --- | --- |
| GPT4o+SoM [37] | – | – | 18.9 |
| ICAL (ours) | 32.3 | 22.3 | **23.4** |
| GPT4V+SoM [37] | 16.3 | 14.1 | 14.3 |
| ICAL (ours) | **38.8** | **20.9** | **22.7** |
| *Ablations* | | | |
| GPT4V+SoM [37] | 11.5 | 12.9 | 12.7 |
| ICAL (ours) | 28.0 | 21.6 | 22.2 |
| w/o image | 28.0 | 17.3 | 19.0 |
| w/ full text trajectory | 57.7 | 21.6 | 25.5 |

Table 3: **Evaluation on the Ego4D unseen validation subset.** ICAL outperforms few-shot GPT4V and matches supervised baselines using 639x less in-domain data.

|  | ED@(Z=20) | | |
| --- | --- | --- | --- |
|  | Verb | Noun | Action |
| Supervised [28] (639x more data) | 0.7251 | 0.7393 | 0.9235 |
| Zero-shot CoT [39] | 0.8796 | 0.7930 | 0.9639 |
| Few-shot CoT | 0.7877 | 0.7575 | 0.9414 |
| ICAL (ours) | **0.7802** | **0.6934** | **0.9242** |

### 4.3 ICAL obtains state-of-the-art performance on visual web tasks

We evaluate our agent with learned ICAL examples on the VisualWebArena evaluation set. We partition this into episodes 'seen' by our model during learning, and those 'unseen' during learning.

Table 2 presents the results on VisualWebArena. Our model, ICAL, outperforms the previous state-of-the-art [37], which uses GPT4V with few-shot, hand-designed examples and set-of-marks image prompting [84]. ICAL achieves an absolute 8.4% (relative 58.7%) improvement in average success rate over GPT4V and shows a 23.8% relative improvement in average success rate over GPT4o.

### 4.4 ICAL outperforms few-shot VLMs on egocentric video action forecasting

We test ICAL on video action forecasting without using human-in-the-loop abstraction verification due to the passive nature of the task. As shown in Table 3, ICAL demonstrates superior few-shot performance on Ego4D action anticipation compared to hand-written few-shot GPT4V that uses chain of thought [81], improving by 6.4 noun and 1.7 action edit distance. ICAL also remains competitive with the fully supervised baseline [28] in noun and action prediction despite using 639x less in-domain training data. We find GPT4V video processing to have the least improvements for verb action prediction, possibly due to its limited video understanding capabilities.

### 4.5 ICAL shows continual improvement with more examples

ICAL shows continual improvements in TEACh validation unseen success rate with more examples learned, as shown in Figure 5. This is in contrast to the unchanged visual demos used for seeding ICAL learning, which show only marginal improvements. Importantly, throughout learning, ICAL does not need to worry about forgetting previously learned knowledge since the agent is expanding a memory of examples and testing with a frozen VLM via in-context learning. Also noteworthy, our method benefits from even a small amount of examples learned, with an improvement of an absolute 14.7% success rate over zero-shot chain-of-thought [39] prompting and 6.8% over the unchanged demonstrations (with 10x less data) with just 10 abstracted demonstrations, showing the efficiency of our method.

### 4.6 Example retrieval improves learning efficiency

Efficient learning systems benefit greatly from leveraging past knowledge, allowing them to reduce the need for human intervention and environment interactions as they continue to process new data. Our agent becomes increasingly efficient over time, requiring less human feedback and fewer environment interactions as it processes more examples. By retrieving past successful abstractions during the VLM-abstraction making and human-in-the-loop phases, it uses previously stored knowledge to help abstract new examples. As shown in Figure 6, for the second half of examples processed, the model requires significantly fewer environment steps (436±88 vs. 267±43, p=0.0143) and human feedbacks (0.74±0.17 vs. 0.21±0.08, p=0.0089) per example. This demonstrates that retrieving abstracted examples during abstraction

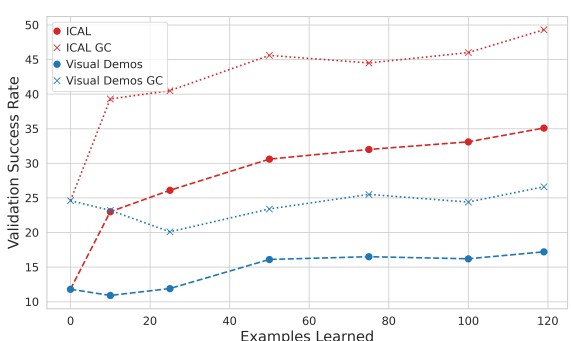

Figure 5: **TEACh validation unseen success rate** for ICAL with increasing number of exemplars. ICAL continually learns without forgetting, significantly outperforming the unchanged visual demos used to seed ICAL learning. ● denotes task success, while **x** denotes goal-condition success.

learning reduces both human effort and environment interaction over time. Consequently, using previously stored ICAL examples not only improves test performance but also accelerates learning for future examples.

### 4.7 Fine-tuning helps

We finetune the GPT3.5-turbo-1106 model on the learned ICAL examples in TEACh using LoRA [32] in the AzureAI interface (see the Appendix Section S5.4 for details). The training data include the 122 successfully abstracted examples by ICAL, which we randomly split into 99 training samples and 23 validation samples. This leads to an improvement of 11.4% task success and 15.7% goal-condition success for the GPT3.5 model. Combining the finetuned model with retrieval-augmented generation using the ICAL examples led to an additional improvement of 0.7% task success and 4.9% goal-condition success over using retrieval-augmented generation without finetuning: our top-performing agent. This demonstrates that consolidating the ICAL learned abstractions with weight fine-tuning helps performance.

### 4.8 Ablations
### show each component of ICAL is important

We ablate the components of ICAL in TEACh in Table 1. We conclude:

1. The abstraction phase significantly helps for refining the trajectories and adding generalizable knowledge. We observe a decrease in 5.7% success rate and 4.4% in goal condition success rate when removing the abstraction phase.

2. The human-in-the-loop phase is important for fixing errors and incorporating feedback from the user. We observe a decrease in 5.2% success rate and 8.3% in goal condition success rate when removing the human-in-the-loop phase.

3. Our examples demonstrate scalability with larger LLMs. GPT-4 showed a 6.6% absolute increase in task success and a 14.3% absolute rise in goal condition success compared to GPT-3.5.

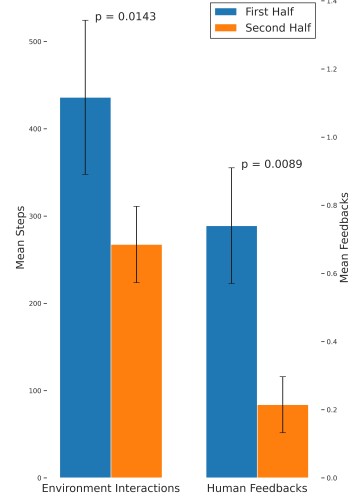

Figure 6: **ICAL improves learning efficiency as more examples are added to memory.** First half (blue) versus second half (orange) of ICAL learning across tasks **(left)** and for each task type separately **(right)** in TEACh. The second half of ICAL learning requires significantly fewer environment steps (436±88 vs. 267±43, p=0.0143) and human feedbacks per episode (0.74±0.17 vs. 0.21±0.08, p=0.0089). This indicates that retrieving ICAL examples during learning is beneficial, reducing both human effort and environment interaction over time.

4. ICAL can be combined with advanced prompting and sampling methods. We test this using re-ranking [78], where the model generates three diverse outputs from different retrieved examples (e.g., top 1-5, 6-10, ...), self-evaluates, and selects the highest scoring output. Improvements are modest but notable: 0.2% in task success and 2.5% in goal condition success.

## 5 Conclusion

We presented ICAL, a method that improves in-context learning by learning to abstract noisy demonstrations into actionable insightful plans, that when used as in-context examples improve performance of VLM agents over in-context learning from raw examples. ICAL proposes abstracting in-context examples as a general form of quick learning from a handful of demonstrations and human-feedback. It also reduces the need for expert examples, and enables more efficient learning. Tested in TEACh, VisualWebArena, and Ego4D, ICAL achieves state-of-the-art performance, demonstrating adaptability to new tasks and environments. There are several limitations and future research directions for ICAL. While ICAL can handle noisy demos, ICAL may not be able to handle extremely misleading demonstrations or feedback, and relies on a fixed action API which may restrict adaptability in dynamic environments. Additionally, GPT4V's visual grounding deficiencies [90, 82, 55, 9] cannot always be overcome by in-context learning, and more research is needed to address this.

**Acknowledgements** This material is based upon work supported by National Science Foundation grants GRF DGE1745016 & DGE2140739 (GS), ONR award N00014-23-1-2415, AFOSR Grant FA9550-23-1-0257, and DARPA No. HR00112490375 from the U.S. DARPA Friction for Accountability in Conversational Transactions (FACT) program. Any opinions, findings and conclusions or recommendations expressed in this material are those of the authors and do not necessarily reflect the views of the United States Army, the National Science Foundation, or the United States Air Force.

This research project has benefitted from the Microsoft Accelerate Foundation Models Research (AFMR) grant program through which leading foundation models hosted by Microsoft Azure along with access to Azure credits were provided to conduct the research.

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

## S1  Overview

The structure of this Appendix is as follows:

- Section S2 contains negative potential impacts.
- Section S4 contains additional experiments.
- Section S5 contains additional methods details.
- Section S6 contains additional details on the evaluation environments.

## S2  Potential Negative Impact

The work introduced by ICAL for AI agents carries potential risks including the perpetuation of biases, privacy infringement, user dependency, economic displacement, security vulnerabilities, and the emergence of unintended behaviors due to technical limitations. Mitigating these risks necessitates the development of mechanisms for bias correction, privacy preservation, ethical guidelines, and security protocols. Engaging with a diverse range of stakeholders is imperative to ensure that the deployment of these technologies aligns with societal values and contributes positively to the realm of human-AI collaboration, fostering advancements that are both innovative and responsible.

## S3  ICAL relation to dAgger

### S3.0.1  Relation of Abstraction Verification to DAgger

The human-in-the-loop phase of ICAL bears a conceptual resemblance to the Dataset Aggregation (DAgger) algorithm [67], as both methods involve iterative refinement of an agent's policy through interaction with expert feedback. However, ICAL extends this framework by incorporating natural language feedback, updating both actions and abstractions, and utilizing retrieval-augmented generation (RAG) with an explicit memory of optimized examples for policy improvement.

In DAgger, the agent collects data by executing its current policy and then queries an expert to obtain the correct action for each encountered state. Specifically, at iteration $t$, the agent observes a state $s_t$ and takes an action $a_t = \pi_t(s_t)$ according to its policy $\pi_t$. The expert provides the optimal action $a_t^*$, and the agent aggregates this data into a dataset $\mathcal{D}$:

$$\mathcal{D} = \mathcal{D} \cup \{(s_t, a_t^*)\}. \tag{5}$$

The policy is then updated by minimizing a loss function over $\mathcal{D}$:

$$\pi_{t+1} = \arg\min_{\pi} \sum_{(s_i, a_i^*) \in \mathcal{D}} L(\pi(s_i), a_i^*). \tag{6}$$

Similarly, in ICAL's human-in-the-loop phase $F_{\text{hitl}}$, the agent refines its behavior based on human feedback. When the agent executes an optimized trajectory $\xi_{\text{optimized}}$ and encounters a failure at action $a_t$, a human observer provides natural language feedback $H(a_t, o_t)$ concerning the action $a_t$ and the observation $o_t$. The agent integrates this feedback to update both the trajectory and the associated language abstractions:

$$(\xi'_{\text{optimized}}, L') = \Xi_{\text{update}}(\xi_{\text{optimized}}, H(a_t, o_t), L, I, \{e^1, \ldots, e^k\}), \tag{7}$$

where $L$ represents the current language annotations, $I$ is the task instruction, and $\{e^1, \ldots, e^k\}$ are retrieved examples from memory. This updated trajectory $\xi'_{\text{optimized}}$ and abstractions $L'$ are then added to an explicit memory $\mathcal{E}$, enhancing the agent's policy through enriched context:

$$\mathcal{E} = \mathcal{E} \cup \{(\xi'_{\text{optimized}}, L')\}. \tag{8}$$

The agent's policy $\pi_{\text{ICAL}}$ is implicitly updated by conditioning on this memory during action generation:

$$\pi_{\text{ICAL}}(s_t, \mathcal{E}) = \text{VLM}(s_t, \mathcal{E}), \tag{9}$$

where VLM denotes the Vision-Language Model used for in-context learning.

The similarities between ICAL and DAgger lie in their iterative approach to policy refinement using expert feedback. However, ICAL offers several key benefits:

**Natural Language Feedback:** Unlike DAgger, which requires the expert to provide explicit action corrections $a_t^*$, ICAL accepts natural language feedback $H(a_t, o_t)$. This allows the human to convey richer information, including explanations, suggestions, and contextual details that can address not only the immediate failure but also underlying misconceptions.

**Revision of Actions and Abstractions:** ICAL updates both the action sequence and the associated language abstractions $L$. By refining the abstractions, the agent enhances its understanding of task structures, causal relationships, and state changes, which promotes better generalization to new tasks and environments.

**Policy Improvement via Retrieval-Augmented Generation:** ICAL maintains an explicit memory $\mathcal{E}$ of optimized examples and abstractions. During deployment, the agent retrieves relevant examples from $\mathcal{E}$ based on similarity measures and uses them as context for action generation. This retrieval-augmented generation (RAG) approach allows the agent to leverage past experiences effectively, adapting its policy without explicit parameter updates.

In contrast, DAgger relies solely on aggregating state-action pairs and updating the policy through supervised learning, which may not capture higher-level task structures or facilitate transfer to new domains. ICAL's ability to process natural language feedback and to update both actions and abstractions provides a more flexible and powerful framework for policy refinement, aligning more closely with human learning processes. ICAL extends the traditional imitation learning paradigm represented by DAgger, enabling more efficient and generalizable learning from human feedback.

## S4 Additional Experiments

### S4.1 Learning efficiency broken down by task type

In Section 4.6 of the main paper, we showed how ICAL enables fewer environment interactions and human feedbacks per example. We provide the learning efficiency between the first and second half of demonstrations processed broken down by task type in Figure S1.

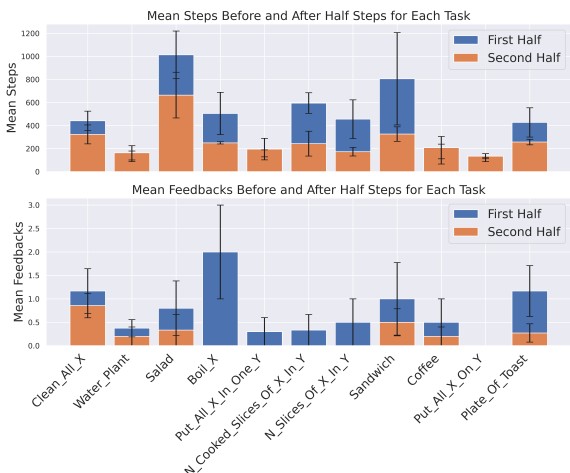

Figure S1: First half (blue) versus second half (orange) of ICAL learning across tasks **(left)** and for each task type separately **(right)** in TEACh.

### S4.2 Experimenting with different types of in-context examples in VisualWebArena

We experiment with an alternate way to provide the ICAL in-context examples to the VLM. Instead of retrieving a single time step, we give the full trajectory of observations, abstractions, and actions in textual format (no images provided). We run this on a reduced subset of 239 VisualWebArena

Table S1: VisualWebArena performance for ICAL performance when using a single time step with image input for in-context examples versus providing all time steps, but without image inputs.

|  | Seen | Unseen | Average |
|---|---|---|---|
| GPT4V+SoM [37] | 11.5 | 12.9 | 12.7 |
| ICAL + text + full trajectory | **57.7** | **21.6** | **25.5** |
| ICAL + text + single time step | 28.0 | 17.3 | 19.0 |
| ICAL + text + single time step + image | 28.0 | **21.6** | 22.2 |

Table S2: Tasks successfully completed after applying the ICAL method (out of 250). We compare ICAL using either visual demonstrations or kinesthetic demonstrations. Kinesth. = Kinesthetic; demos with GT actions. Visual = action labeled from RGB frames with inverse dynamics model.

|  | GPT-4 | | GPT-3.5 |
|---|---|---|---|
|  | Visual Demos | Kinesth. Demos | Visual Demos |
| Breakfast | 0 | 7 | 0 |
| Boil_X | 5 | 11 | 3 |
| Water_Plant | 13 | 16 | 9 |
| Salad | 11 | 8 | 2 |
| Sandwich | 9 | 8 | 0 |
| Put_All_X_On_Y | 12 | 15 | 6 |
| Plate_Of_Toast | 17 | 14 | 9 |
| N_Slices_Of_X_In_Y | 11 | 13 | 6 |
| Clean_All_X | 13 | 16 | 3 |
| Put_All_X_In_One_Y | 13 | 16 | 10 |
| Coffee | 9 | 15 | 1 |
| N_Cooked_Slices_Of_X_In_Y | 5 | 3 | 3 |
| **Total** | 122 | 142 | 52 |

tasks. The results are presented in Table S1. We find that providing the full trajectory increases seen success rate, but does not improve unseen success rate. In our final evaluation, we utilize the retrieval of a single time step with image input, since expanding the context length through the full trajectory adds to the cost without significantly improving the success rate on unseen tasks.

### S4.3 TEACh results on ICAL learning using trajectories with ground truth action labels and GPT3.5

We present tasks successfully completed for each task type in Table S2, comparing ICAL that uses visual demonstrations and kinesthetic demonstrations.

We see that using GPT3.5 for ICAL significantly reduces the number of successful tasks by over half compared to using GPT4 (52 versus 122 tasks successfully completed). We show in Section S4.5 of the main paper how relabeling unsuccessful tasks can help improve performance when using weaker models, such as GPT 3.5.

### S4.4 TEACh validation accuracy by task type

We present ICAL agent performance after learning on the TEACh validation set for each task type in Table S3.

### S4.5 Relabeling unsuccessful examples improves ICAL when using weaker models

Instead of removing unsuccessful examples, we can instead relabel the examples by querying an LLM to generate a new task instruction, step-by-step plan, and summary for the partial task completion. In Table S4, we show performance of ICAL with and without relabeling. Relabeling improves performance when using a weaker model, GPT3.5 during ICAL learning, by an absolute 3.4% in success and 0.5% in GC.

Table S3: TEACh validation performance of ICAL after examples have been learned for each task type when evaluated using GPT3.5 or GPT4.

|  | GPT3.5 | | GPT4 | |
| --- | --- | --- | --- | --- |
|  | Success (%) | GC (%) | Success (%) | GC (%) |
| Breakfast | **2.3** | 37.5 | **2.3** | **56.6** |
| Boil_X | **18.2** | **18.2** | 13.6 | 13.6 |
| Water_Plant | **73.0** | **73.0** | 61.9 | 61.9 |
| Salad | 31.9 | 61.7 | **40.4** | **74.6** |
| Sandwich | 2.1 | 45.4 | **2.1** | **56.0** |
| Put_All_X_On_Y | 56.0 | 65.6 | **66.0** | **74.0** |
| Plate_Of_Toast | 0.0 | 50.0 | **11.7** | **60.2** |
| N_Slices_Of_X_In_Y | 48.1 | 57.6 | **63.0** | **72.2** |
| Clean_All_X | 58.6 | 58.8 | **70.7** | **75.0** |
| Put_All_X_In_One_Y | 62.5 | 72.0 | **66.7** | **72.0** |
| Coffee | 31.6 | 44.6 | **49.1** | **59.5** |
| N_Cooked_Slices_Of_X_In_Y | 16.3 | 44.7 | **30.6** | **70.0** |
| **Average** | 35.1 | 49.3 | **41.7** | **63.6** |

Table S4: Relabeling experiments. Relabeling unsuccessful demonstrations improves performance when using weaker models, such as GPT3.5, during the ICAL learning process.

|  | Success | GC |
| --- | --- | --- |
| ICAL | 22.4 | 36.9 |
| + relabeling | **25.8** | **37.4** |

## S4.6 Running ICAL from RGB-only input

We run ICAL from RGB inputs. We use the perception, navigation, and manipulation modules from HELPER [68], which uses SOLQ [20] for object detection and ZeoDepth [5] for depth estimation. HELPER initializes objects with default attributes based on the domain, and uses domain-specific pre-condition checks and error correction. However, we wish to automate the learning of these modules, and thus remove them. Additionally, HELPER initializes a memory of examples hand-written by a domain expert. We replace these with our ICAL examples. For inferring attributes of the objects in the abstracted state, we apply CogVLM [77], an open-source visual language model, on the detected object crops, which we found to work best compared to other models for object attribute detection on a separate dataset of cropped object images (see Appendix).

As shown in Table S5, we find that our ICAL agent obtains performance close to that of HELPER, lagging behind 1.7% success and 3.2% goal condition success, despite not hand-designing object attributes, pre-condition checks, error correction, and in-context examples. Additionally, when using the hand-written HELPER examples with the ICAL execution modules, we find that the ICAL examples outperform the HELPER examples by 2.2% in task success and by 1.3% in goal-condition success, despite the ICAL examples being obtained without hand-writing from a domain-expert. Additionally, when using the perception of ODIN [33], which utilizes multi-view images and a 3D bottleneck for semantic segmentation, ICAL obtains performance on-par with that of HELPER.

## S4.7 Benchmarking open-source VLMs for attribute detection in TEACh

In household instruction following, ICAL benefits from accurate object and attribute detection from sensory input for state inference. For benchmarking object attribute detection in TEACh, we build a dataset of 2581 object crops of clean viewpoints of the object by having the agent pick up the object and centering the object in view. We build a second dataset of 661 from random viewpoints of the object in the TEACh training set with different objects varying in their "dirty" and "cooked" attributes. Clean viewpoints are always centered, unoccluded, and posed, while the random viewpoints are often occluded and show the object from different angles. Example crops for the datasets for a 'dirty plate' is shown in Figure S2. We test the following models: OPENCLIP CLIP-VIT-BIGG-14-LAION2B-39B-B160K[13], OpenAI CLIP *clip-vit-base-patch32*[65], *X-VLM*[87], *Llava 1.5*[48], *cogVLM*[77],

Table S5: TEACh validation set (seen) from RGB input. Our ICAL agent obtains performance close to that of HELPER, despite not hand-designing object attributes, pre-condition checks, error correction, and in-context examples.

|  | Success | GC |
|---|---|---|
| E.T. [64] | 1.0 | 1.4 |
| JARVIS [91] | 1.7 | 5.4 |
| FILM [57] | 5.5 | 5.8 |
| DANLI [89] | 5.0 | 10.5 |
| HELPER [68] | 12.2 | 18.6 |
| ICAL w/ HELPER examples | 8.3 | 14.1 |
| ICAL (ours) | 10.5 | 15.4 |
| HELPER [68] + ODIN [33] | 13.8 | 26.6 |
| ICAL (ours) + ODIN | 13.8 | 25.5 |

Table S6: Attribute detection accuracy in TEACh for different open-source VLMs. We find CogVLM currently outperforms the other open-source VLMs at posed and unposed attribute detection for object crops.

|  | Clean Views | Random Views |
|---|---|---|
| OpenCLIP CLIP-ViT-bigG-14-laion2B-39B-b160k [13] | 0.860 | 0.715 |
| OpenAI CLIP clip-vit-base-patch32 [65] | 0.758 | – |
| X-VLM [87] | 0.785 | – |
| Llava 1.5 [48] | 0.862 | 0.839 |
| cogVLM [77] | **0.898** | **0.857** |
| Diffusion Classifier [43] | 0.665 | – |
| Open Flamingo [2] | 0.530 | – |

*diffusion classifier*[43], *Open Flamingo*[2]. We queried CLIP by taking the best match of the image encoding with [a photo of a {category} that is {word1}, a photo of a {category} that is {word2}], where word1 and word2 are opposite attributes. We queried diffusion classifier with [a blurry photo of a {word1} {category}., a blurry photo of a {word2} {category}.], as per the paper. We queried CogVLM, Llava, and Open Flamingo with the image crop and asked it Is this {category} {word1} or {word2}? Provide a single word answer, either "{word1}" or "{word2}". We show the results on our evaluation dataset in Table S6. We find CogVLM outperforms the other open-source VLMs at posed and unposed attribute detection for object crops. We use cogVLM for our estimated perception experiments for detecting object attributes.

## S5 Additional Methods Details

### S5.1 In-Context Abstraction Learning (ICAL) Algorithm

We present the method for In-Context Abstraction Learning (ICAL) for a single trajectory in Algorithm S1. Given a noisy trajectory, the method proceeds by first abstracting the trajectory through a function $F_{abstract}$, which leverages a LLM or VLM to correct errors or inefficiencies in the trajectory and generates language abstractions that capture the essence of the task, including subgoals, causal relationships, and state changes. This phase does not require interaction with the environment or humans.

Initialization sets up for the Human-In-The-Loop (HITL) phase by resetting the feedback count and success flag. The method then enters a feedback loop where the optimized trajectory is executed in the environment. If the task execution is successful, the loop breaks, and the method proceeds to update the example set with the abstracted trajectory and its associated language abstractions. Otherwise, human feedback is solicited at the point of failure to revise the trajectory and language

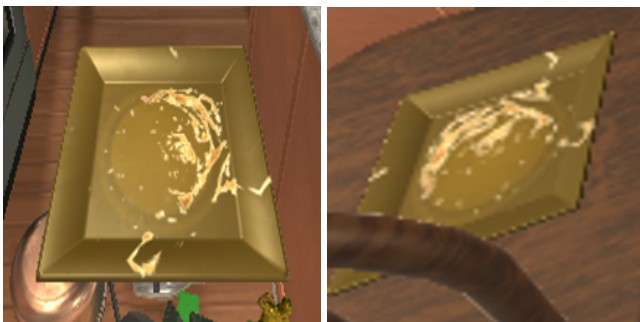

Figure S2: An example image crop of the attribute dataset collected in TEACh of a dirty plate. A clean viewpoint example image is shown on the left, and a random viewpoint example image is shown on the right.

abstractions further, utilizing the VLM again. This feedback loop continues until either the task is successfully executed or a predefined maximum number of feedback iterations is reached.

### S5.1.1 Noisy Trajectories

We collect a noisy sequence of observations and actions, denoted as $\xi_{noisy} = o_0, a_0, \ldots, o_T, a_T$, which represents a trajectory for the language-defined task we aim for our agent to learn and adapt to.

The trajectory sequences can come from a variety of sources, including unlabeled video sequences. We can also accommodate sub-optimal or unsuccessful attempts. In our work, we identify three scenarios in which a given sequence, $\xi_{noisy}$, might be inefficient or incorrect, and using them directly as in-context examples for our LLM/VLM agents could result in poor performance:

- **Human Non-experts:** We gather demonstrations from humans without domain-specific expertise. Specifically, these demonstrations $o_0, a_0, \ldots, o_T, a_T$ are collected from human participants who are provided with a textual instruction $I$ and an RGB image at each time step, and are instructed to choose actions $a \in A$ to complete the instruction. These humans commit errors, choose sub-optimal actions, and may not complete the task to its fullest extent. For instance, an episode within TEACh [63] has a participant who picks up a knife and then looks up and down before placing the knife down again, a sub-optimal action sequence not required by the instruction [58].

- **Visual Passive Demonstrations:** Here, the agent is given a sequence of observations $\xi = \{o_0, o_1, \ldots, o_T\}$ that lack corresponding action labels. We collect these visual demonstrations for TEACh only. We use the TEACh human demonstrations as described in the previous text and take the egocentric RGB images without actions as the observation sequence. To infer the actions executed in these demonstrations, an inverse dynamics model is applied to consecutive pair of frames $F_{idm}(o_t, o_{t+1})$, which predicts the action $a_t$ responsible for the state transition $o_t \rightarrow o_{t+1}$. Along with sub-optimal human trajectories, the inverse dynamics may make prediction errors. We trained a transformer encoder-decoder model based on the DETR [7] architecture on 300 TEACh episodes (see Section S5.3 for more details).

- **Agent Trajectories:** In Visual Web Arena [37], we obtain additional demonstrations sourced from deploying our in-context VLM on new tasks. Specifically, we first run our ICAL process on 30 human demonstrations collected by non-experts. We run the ICAL process to abstract the 30 human demonstrations and then deploy our ICAL agent using the learned examples as in-context examples. Using this ICAL agent, we collect an additional 62 new trajectories on Visual Web Arena tasks and continue to run the ICAL learning on these new trajectories collected by the model.

### S5.1.2 Abstraction phase implementation details

We present our prompt template for the VLM abstraction generation phase in Listing S5.

**TEACh.** We iterate through each Python program demonstration labeled with the inverse dynamics model. Given the noisy Python program, instruction, action API, and object state, the abstraction phase proceeds by prompting the LLM to 1) revise the the code for maximal efficiency and fix mistakes in the code (abstracted trajectory), 2) provide a summary of the functionality of the script (Task Decomposition & Subgoals), 3) provide a step-by-step plan of the steps of the script (Task Decomposition & Subgoals), 4) Add object attribute state changes to the Python program (State Changes), and 5) add abstraction comments (Task and Causal Abstractions). For state changes, we parameterize the state changes in TEACh, allowing the LLM to add a change_state() function to the actions to indicate a change in state of the objects the agent is interacting with. Each step uses retrieved examples of successful examples previously saved in memory.

**VisualWebArena.** We perform the abstraction phase for each individual action taken $a_t$ (e.g., click(element), hover(element)) in each noisy trajectory $\xi_{noisy}$ obtained in VisualWebArena. Specifically, for each action in $\xi_{noisy}$, we first prompt the VLM to optionally revise the action (optimized trajectory), and output a summary and step-by-step reasoning for the chosen action (Task Decomposition & Subgoals), given the instruction, image observation, textual state description, previous actions taken, and proposed trajectory action for the current time step. Next, we prompt the VLM to output a a predicted next state (State Changes), given the instruction, current and next image observation, current and next textual state, and the action taken $a_t$. We next prompt the VLM to output the most relevant state elements for the task instruction (State Abstraction), given the instruction, image observation, and textual state.

Finally, we prompt the VLM to output a set of abstraction comments, given the full sequence of abstracted actions, task decomposition and subgoals, state changes, and abstracted state descriptions.

**Ego4D.** We perform the abstraction phase for each full demonstration, consisting of 20 predicted future times steps of actions. We give the VLM 3 video frames annotated with set-of-marks, the GT actions, and the actions in the video, and prompt the VLM to annotate the four types of abstractions for each example.

### S5.1.3 Human-in-the-loop phase implementation details

We present our prompt template for the human-in-the-loop phase revisions in Listing S6.

**TEACh.** The TEACh simulator enables fine-grained analysis of task progress. During the human-in-the-loop phase, we formulate this task progress into natural language feedback for failed actions (e.g., "The Toaster is full right now." or missed task steps (e.g., "The Pillow needs to be put onto a Sofa"). The natural language feedback, along with the instruction, object state, action API, and failed actions/code, are given to the LLM to revise the program and abstractions.

**VisualWebArena.** For 20 tasks for each website related to the tasks in the demonstrations collected, we develop an interface to provide natural language corrections to the model based on the predicted next action by the model. The humans are tasked to intervene and provide feedback when they deem an action predicted by the model sub-optimal. When an action is proposed, the humans are given the ability to accept the action or reject the action if it is sub-optimal. If sub-optimal, the humans can type in natural language feedback which will be sent to the VLM to revise the action and abstractions. We provide an example of the previous outputs, human feedback, and revised outputs in Listing S1.

**Ego4D.** Due to the passive nature of the Ego4D task, where there is no agent executing the actions predicted, no human in the loop phase is implemented. Ego4D only uses the abstraction phase.

**Algorithm S1** In-Context Abstraction Learning (ICAL) method for a single trajectory

---

**Require:** Noisy trajectory $\xi_{noisy} = \{o_0, a_0, ..., o_T, a_T\}$, Task instruction $I$, Maximum feedback iterations $N_{feedbacks}$
**Ensure:** Updated example set M

1: **Abstraction Phase:**                                                                                 ▷ Abstraction of the trajectory
2: $(\xi_{abstracted}, L) \leftarrow F_{abstract}(\xi_{noisy}, I, \{e^1, ..., e^k\})$                   ▷ Prompts VLM

3: **Initialization:**                                                                                    ▷ Prepare for the HITL phase
4: $feedback\_count \leftarrow 0$
5: $success \leftarrow$ False

6: **while** $feedback\_count < N_{feedbacks}$ **and not** $success$ **do**                              ▷ Feedback loop
7:     Execute $\xi_{abstracted}$ in the environment                                                     ▷ Attempt task execution
8:     **if** Task is successful **then**
9:         $success \leftarrow$ True
10:         **break**
11:     **else**
12:         Obtain human feedback $H$                                                                     ▷ Get feedback at failure point
13:         $(\xi_{abstracted}, L) \leftarrow \Xi_{update}(\xi_{abstracted}, H, L, I, \{e^1, ..., e^k\})$  ▷ Prompts VLM
14:         $feedback\_count \leftarrow feedback\_count + 1$
15:     **end if**
16: **end while**

17: **if** $success$ **then**
18:     Update example set $M \leftarrow M \cup \{(\xi_{abstracted}, L)\}$                                ▷ Store the abstracted example
19: **end if**

---

Listing S1: Example of human-in-the-loop revision for visual web arena.

```
Instruction: Show me the most recently posted blue iPhone.

-------------------------------------------------------------------------------------

Abstractions & action before human feedback:

Abstraction Comments:
- When searching for the most recently posted item of a specific color, ensure the
    listings are sorted by the newest first and visually inspect the images for the
    color before clicking on the correct listing.

Plan: To find the most recently posted blue iPhone, I need to use the search function on
    the classifieds website. The search input field is identified by [5]. I will type "
    blue iPhone" into the search box to filter the listings for items related to blue
    iPhones. This will help me find the most recently listed blue iPhone.

Summary: Typing "blue iPhone" into the search field will filter the listings to show
    relevant items, allowing progress towards identifying the most recently posted blue
    iPhone.

Predicted Next State: After typing "blue iPhone" into the search field and initiating the
    search, the next state will display search results related to blue iPhones,
    potentially including listings with images and descriptions of the items.

Action: In summary, the next action I will perform is ```type [5] [blue iPhone]```

-------------------------------------------------------------------------------------

Human feedback: "That search likely will not work as well as clicking on the Cell phones
    button and searching for the iPhone there on this classifieds site."

-------------------------------------------------------------------------------------

Abstractions & action before human feedback:

Abstraction Comments:
- When searching for the most recently posted item of a specific color, ensure the
    listings are sorted by the newest first and visually inspect the images for the
    color before clicking on the correct listing.
- When searching for specific items on a classifieds website, it is more effective to
    narrow down the search by selecting the appropriate category before typing in the
    search term.

Plan: Let's think step-by-step. The objective is to find the most recently posted blue
    iPhone. The feedback indicates that I should first narrow down the category by
    clicking on the 'Cell phones' category, which has the ID [41]. After selecting the
    category, I can then search for 'blue iPhone' within that category.

Summary: To find the most recently posted blue iPhone, I need to first narrow down the
    search by selecting the 'Cell phones' category.

Predicted State: After clicking on the 'Cell phones' category, the page will display
    listings under that category, allowing for a more targeted search for a blue iPhone.

Revised Action: In summary, the next action I will perform is ```click [41]```
```

### S5.1.4  Relation of Abstraction Verification to DAgger

The human-in-the-loop phase of ICAL bears a conceptual resemblance to the Dataset Aggregation (DAgger) algorithm [67], as both methods involve iterative refinement of an agent's policy through interaction with expert feedback. However, ICAL extends this framework by incorporating natural language feedback, updating both actions and abstractions, and utilizing retrieval-augmented generation (RAG) with an explicit memory of optimized examples for policy improvement.

In DAgger, the agent collects data by executing its current policy and then queries an expert to obtain the correct action for each encountered state. Specifically, at iteration $t$, the agent observes a state $s_t$ and takes an action $a_t = \pi_t(s_t)$ according to its policy $\pi_t$. The expert provides the optimal action $a_t^*$, and the agent aggregates this data into a dataset $\mathcal{D}$:

$$\mathcal{D} = \mathcal{D} \cup \{(s_t, a_t^*)\}. \tag{10}$$

The policy is then updated by minimizing a loss function over $\mathcal{D}$:

$$\pi_{t+1} = \arg\min_{\pi} \sum_{(s_i, a_i^*) \in \mathcal{D}} L(\pi(s_i), a_i^*). \tag{11}$$

Similarly, in ICAL's human-in-the-loop phase $F_{\text{hitl}}$, the agent refines its behavior based on human feedback. When the agent executes an optimized trajectory $\xi_{\text{optimized}}$ and encounters a failure at action $a_t$, a human observer provides natural language feedback $H(a_t, o_t)$ concerning the action $a_t$ and the observation $o_t$. The agent integrates this feedback to update both the trajectory and the associated language abstractions:

$$(\xi'_{\text{optimized}}, L') = \Xi_{\text{update}}(\xi_{\text{optimized}}, H(a_t, o_t), L, I, \{e^1, \ldots, e^k\}), \tag{12}$$

where $L$ represents the current language annotations, $I$ is the task instruction, and $\{e^1, \ldots, e^k\}$ are retrieved examples from memory. This updated trajectory $\xi'_{\text{optimized}}$ and abstractions $L'$ are then added to an explicit memory $\mathcal{E}$, enhancing the agent's policy through enriched context:

$$\mathcal{E} = \mathcal{E} \cup \{(\xi'_{\text{optimized}}, L')\}. \tag{13}$$

The agent's policy $\pi_{\text{ICAL}}$ is implicitly updated by conditioning on this memory during action generation:

$$\pi_{\text{ICAL}}(s_t, \mathcal{E}) = \text{VLM}(s_t, \mathcal{E}), \tag{14}$$

where VLM denotes the Vision-Language Model used for in-context learning.

The similarities between ICAL and DAgger lie in their iterative approach to policy refinement using expert feedback. However, ICAL offers several key benefits:

**Natural Language Feedback:** Unlike DAgger, which requires the expert to provide explicit action corrections $a_t^*$, ICAL accepts natural language feedback $H(a_t, o_t)$. This allows the human to convey richer information, including explanations, suggestions, and contextual details that can address not only the immediate failure but also underlying misconceptions.

**Revision of Actions and Abstractions:** ICAL updates both the action sequence and the associated language abstractions $L$. By refining the abstractions, the agent enhances its understanding of task structures, causal relationships, and state changes, which promotes better generalization to new tasks and environments.

**Policy Improvement via Retrieval-Augmented Generation:** ICAL maintains an explicit memory $\mathcal{E}$ of optimized examples and abstractions. During deployment, the agent retrieves relevant examples from $\mathcal{E}$ based on similarity measures and uses them as context for action generation. This retrieval-augmented generation (RAG) approach allows the agent to leverage past experiences effectively, adapting its policy without explicit parameter updates.

In contrast, DAgger relies solely on aggregating state-action pairs and updating the policy through supervised learning, which may not capture higher-level task structures or facilitate transfer to new domains. ICAL's ability to process natural language feedback and to update both actions and abstractions provides a more flexible and powerful framework for policy refinement, aligning more closely with human learning processes. ICAL extends the traditional imitation learning paradigm represented by DAgger, enabling more efficient and generalizable learning from human feedback.

### S5.2   Deploying the ICAL agent after the examples have been learned.

We present our algorithm for deploying our ICAL agent after the examples have been learned on new instructions in Algorithm S2. We additional present our prompt template for the VLM planning after examples have been learned in Listing S7.

### S5.3   Inverse Dynamics Model

In this section, we provide implementation details for the inverse dynamics model used in TEACh. A high-level architecture diagram is shown in Figure S3

*Backbone.* Given an input image pair $x_{\text{images}} \in \mathbb{R}^{2 \times 3 \times H_0 \times W_0}$ (2 frames and 3 color channels), we use a CNN to produce lower-resolution activation maps $f \in \mathbb{R}^{2 \times C \times H \times W}$, where $H_0$ and $W_0$ denote the original height and width, respectively, and $H$ and $W$ represent the dimensions of the resulting feature map.

*Transformer Encoder.* The spatial features are input into a transformer encoder, where they undergo self-attention. We reshape the spatial dimensions into a single dimension, resulting in a feature

**Algorithm S2** Deploying the ICAL agent after the examples have been learned.

**Require:**
 Predefined action API based on skill set $A$.
 Set of in-context examples $M = \{e^1, e^2, \ldots, e^k\}$.
 Language instruction $I$.
 Initial observation $o_0$.
 Initial textual observation $x_0$.
 Maximum steps $T$.
1: Initialize observation $o_t \leftarrow o_0$, $x_t \leftarrow x_0$.
2: **for** $t = 0$ to $T - 1$ **do**
3:  Retrieve top $K$ examples: $\{e_t^1, \ldots, e_t^K\} \leftarrow \text{RetrieveTopK}(x_t, o_t, I, M)$.
4:  $a_t \leftarrow \text{VLM}(x_t, o_t, I, \{e_t^1, \ldots, e_t^K\}, A)$ to generate an action or Python code.
5:  Execute action $a_t$ to receive new observation $o_{t+1}, x_{t+1}$.
6:  **if** stop criteria met **then**
7:   **break**
8:  **end if**
9: **end for**

---

map with dimensions $d \times HW$. Each layer of the encoder consists of a multi-head self-attention mechanism and a feed-forward network. Fixed spatial positional encodings and learned frame encodings are added to the inputs at each attention layer. The transformer encoder comprises six self-attention layers, utilizes eight heads, has an embedding size of 384, and contains six encoder layers.

*Transformer Decoder.* The decoder incorporates cross-attention mechanisms for both object and action queries with respect to the encoder features. It processes $N$ queries simultaneously across its layers. The embeddings, comprising action queries and object queries, add learned positional encodings at the input of each attention layer. Each layer of the decoder includes cross-attention from the queries to the encoder features, self-attention among the query features, and a feed-forward network. The transformer decoder consists of six self-attention layers, employs eight heads, has an embedding size of 384, and includes six encoder layers.

After the decoder, we use a feed-forward network to reduce each embedding from dimension $d$ to a scalar value. These scalar values for actions and objects are concatenated, creating final action logits for each action and each object. The model is trained using cross-entropy loss for both actions and objects, such as 'pickup' and 'apple'. Additionally, we introduce an extra query for 'no object' to accommodate actions that do not involve manipulating an object (e.g., move_forward()).

*Dataset.* We use a random subset of 649 training episodes from the TEACh training dataset and 181 validation episodes from the TEACh validation seen dataset, which do not overlap with the episodes used for the ICAL example learning. We use Each episode contains a trajectory of observations and actions $\{o_0, a_0, \ldots, o_T, a_T\}$. We use each pair of observations $(o_t, o_{t+1})$, and the action $a_t$ responsible for the state transition $o_t \rightarrow o_{t+1}$, as training samples for the network.

*Implementation details.* We use a learning rate of $2e - 5$, batch size of 64, a step learning rate scheduler with $\gamma = 0.1$ and step size = 30 epochs. We use early stopping based on validation loss and train for 45 epochs. We use cross entropy loss with a manual class weight rescaling based on frequencies in the training set. Training and model implementation is done in PyTorch.

*Applying Inverse Dynamics Model on held-out demonstrations.* On the held out 250 TEACh episodes used for ICAL example learning, we feed each pair of observations to the trained inverse dynamics model to predict actions. We convert the sequence of predicted actions into a Python program based on the ICAL action API for TEACh. This involves converting each action into a Python function and aggregating contiguous navigation actions (move_forward(), turn_left()) into a single go_to() function in the program. An example of the predicted Python program is shown in Listing S2. We also provide the fully revised program after running ICAL learning on the predicted program in Listing S3.

Listing S2: Demonstration program of actions inferred from the inverse dynamics model for an episode of making a Salad.

```python
target_fridge = InteractionObject("Fridge", object_instance = "Fridge")
target_fridge.go_to()
target_fridge.open()
target_lettuce = InteractionObject("Lettuce", object_instance = "Lettuce")
target_countertop = InteractionObject("CounterTop", object_instance = "CounterTop")
target_lettuce.pickup_and_place(target_countertop)
target_knife = InteractionObject("Knife", object_instance = "Knife")
target_knife.pickup()
target_lettuce.go_to()
target_lettuce.slice()
target_bread = InteractionObject("Bread", object_instance = "Bread")
target_bread.go_to()
target_bread.slice()
target_countertop.go_to()
target_knife.place(target_countertop)
target_tomato = InteractionObject("Tomato", object_instance = "Tomato")
target_tomato.pickup_and_place(target_countertop)
target_knife.pickup()
target_bread.go_to()
target_bread.slice()
target_countertop.go_to()
target_knife.place(target_countertop)
target_breadsliced = InteractionObject("BreadSliced", object_instance = None,
    parent_object = "Bread")
target_plate = InteractionObject("Plate", object_instance = "Plate")
target_breadsliced.pickup_and_place(target_plate)
target_breadsliced.pickup()
target_pot = InteractionObject("Pot", object_instance = "Pot")
target_pot.go_to()
target_breadsliced.place(target_pot)
target_breadsliced.pickup_and_place(target_plate)
target_spoon = InteractionObject("Spoon", object_instance = "Spoon")
target_spoon.pickup()
target_countertop.go_to()
target_spoon.place(target_countertop)
target_lettucesliced = InteractionObject("LettuceSliced", object_instance = None,
    parent_object = "Lettuce") parent
target_lettucesliced.pickup_and_place(target_plate)
target_lettucesliced.pickup()
target_plate.go_to()
target_lettucesliced.place(target_plate)
target_tomatosliced = InteractionObject("TomatoSliced", object_instance = "TomatoSliced")
target_tomatosliced.pickup_and_place(target_countertop)
```

Listing S3: Revised demonstration program (revised from the program in Listing S1) after abstraction phase cleanup and human-in-the-loop for an episode of making a salad.

```python
# Initialize InteractionObject instances for the fridge, lettuce, knife, tomato, and
    plate
fridge = InteractionObject("Fridge", object_instance="Fridge_71")
lettuce = InteractionObject("Lettuce", object_instance="Lettuce_11")  # Lettuce in the
    fridge
knife = InteractionObject("Knife", object_instance="Knife_73")  # Knife on the countertop
tomato = InteractionObject("Tomato", object_instance="Tomato_80")  # Tomato on the table
plate = InteractionObject("Plate", object_instance="Plate_66")  # Plate on the countertop
sink_basin = InteractionObject("SinkBasin", object_instance="SinkBasin_74")  # Sink basin
faucet = InteractionObject("Faucet", object_instance="Faucet_87")  # Faucet

# Go to the fridge and open it to retrieve the lettuce
fridge.go_to()
fridge.open()

# Retrieve two lettuces from the fridge, slice them, and create instances for the sliced
    lettuce
lettuce.go_to()
lettuce.pickup()
knife.go_to()
knife.pickup()
lettuce.slice()
lettuce_sliced_1 = InteractionObject("LettuceSliced", parent_object=lettuce.
    object_instance)  # Initialize new sliced object from sliced parent
lettuce_sliced_2 = InteractionObject("LettuceSliced", parent_object=lettuce.
    object_instance)  # Initialize new sliced object from sliced parent

# Go to the tomato on the table, slice it, and create an instance for the sliced tomato
tomato.go_to()
tomato.slice()
tomato_sliced = InteractionObject("TomatoSliced", parent_object=tomato.object_instance)
    # Initialize new sliced object from sliced parent

# Check if the plate is dirty and clean it if necessary
plate.go_to()
if plate.check_attribute("dirty", True):
    sink_basin.go_to()
    plate.place(sink_basin)
    faucet.go_to()
    faucet.toggle_on()  # Turn on the faucet to clean the plate
    faucet.toggle_off()  # Turn off the faucet after cleaning
    plate.pickup()  # Pick up the clean plate
    plate.change_state("dirty", False)  # Update the plate's state to clean

# Place two slices of lettuce and one slice of tomato on the clean plate
lettuce_sliced_1.go_to()
lettuce_sliced_1.pickup()
plate.go_to()
lettuce_sliced_1.place(plate)
lettuce_sliced_2.pickup()
lettuce_sliced_2.place(plate)
tomato_sliced.go_to()
tomato_sliced.pickup()
tomato_sliced.place(plate)

# Close the fridge after retrieving the items
fridge.go_to()
fridge.close()
```

## S5.4  LLM finetuning details

We use the examples obtained from the ICAL method applied in TEACh, a total of 122 examples. We split the dataset randomly into 99 training samples and 23 validation samples. Input tokens for training consist of each example (instruction, object state, and API) with the prompt template used for zero-shot prompting. Output tokens consist of the Python program with abstraction comments for each example The mean input token length per sample is 3145.17, while the mean output token length per sample is 432.82. We use the Azure OpenAI Service for fine-tuning, which uses the next-token prediction objective and LoRA [32] for parameter-efficient finetuning of gpt-35-turbo-1106.

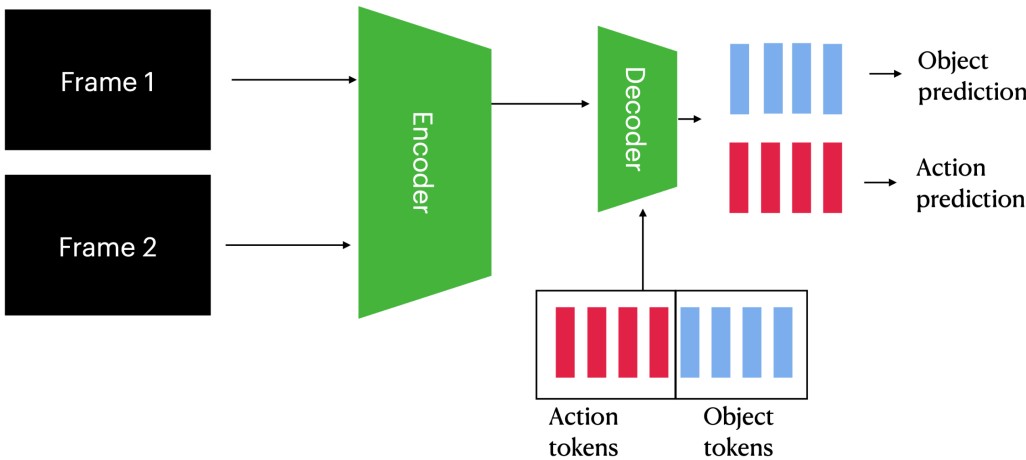

Figure S3: Architecture of inverse dynamics model used for labeling TEACh trajectories.

### S5.5 Compute Resources

We use a single Nvidia RTX A6000 for training the inverse dynamics model and running all our evaluations. We use Azure for finetuning GPT-3.5-1106 as mentioned in Section S5.4. We use Azure OpenAI API for VLM inference.

## S6 Additional implementation details

### S6.1 TEACh

The TEACh dataset builds on the Ai2thor simulation environment [40]. At each time step the agent may choose from the following actions: Forward(), Backward(), Turn Left(), Turn Right(), Look Up(), Look Down(), Strafe Left(), Strafe Right(), Pickup(X), Place(X), Open(X), Close(X), ToggleOn(X), ToggleOff(X), Slice(X), and Pour(X), where X refers an object specified via a relative coordinate $(x, y)$ on the egocentric RGB frame. Navigation actions move the agent in discrete steps. We rotate in the yaw direction by 90 degrees, and rotate in the pitch direction by 30 degrees. The RGB and depth sensors are at a resolution of 480x480, a field of view of 90 degrees, and lie at a height of 0.9015 meters. The agent's coordinates are parameterized by a single $(x, y, z)$ coordinate triplet with $x$ and $z$ corresponding to movement in the horizontal plane and $y$ reserved for the vertical direction. The TEACh benchmark allows a maximum of 1000 steps and 30 API failures per episode.

#### S6.1.1 Planning at test time

Given a new environment and instruction, ICAL first maps out the scene to build a navigation map and detect objects and their attributes (see next sections). ICAL then retrieves the top-$k$ examples relevant to the instruction and object state (see Section 3.4). ICAL then obtains the abstracted object state to give to the LLM (see Section S5.1.2). ICAL then prompts the LLM, given the instruction, abstracted object state, and retrieved in-context examples, to output Python code to carry out the new instruction in the environment. If code execution failures occur, we re-prompt the LLM with the execution error and ask the LLM to revise the code.

#### S6.1.2 ICAL differences with HELPER

In TEACh, we build on HELPER [68] for program execution. Here, we give an account of HELPER. HELPER prompts an LLM, namely GPT-4 [1], to generate plans as Python programs. It assumes that the agent has access to a set of action skills $S$ (e.g., go_to(X), pickup(X), etc.). We use a reduced set of these skills (e.g., we remove the cook(), clean(), and toast() primitives as we wish for our model to learn these). HELPER generates code that is decomposed into these action skills. Instead of

demposing them into action primitives, we run the Python code generated from the LLM directly (i.e., using the 'exec' function in Python). Each action skill comes with a set of pre-engineered pre-condition checks, which we also remove. HELPER maintains a 3D semantic map for navigation and keeping track of objects (see next sections).

### S6.1.3  Obstacle map

ICAL maintains a 2D overhead occupancy map of its environment $\in \mathbb{R}^{H \times W}$ that it updates at each time step from the input RGB-D stream. The map is used for exploration and navigation in the environment. At every time step $t$, we unproject the input depth maps using intrinsic and extrinsic information of the camera to obtain a 3D occupancy map registered to the coordinate frame of the agent, similar to earlier navigation agents [8]. The 2D overhead maps of obstacles and free space are computed by projecting the 3D occupancy along the height direction at multiple height levels and summing. For each input RGB image, we run a SOLQ object segmentor [20] (pretrained on COCO [46] then finetuned on TEACh rooms) to localize each of 116 semantic object categories. For failure detection, we use a simple matching approach from [57] to compare RGB pixel values before and after taking an action. When using ground truth perception, we use ground truth semantic segmentation, depth maps, object attributes, and action failure detection.

### S6.1.4  Object location and state tracking

We maintain an object memory as a list of object detection 3D centroids and their predicted semantic labels $\{[(X, Y, Z)_i, \ell_i \in \{1...N\}], i = 1..K\}$, where $K$ is the number of objects detected thus far. The object centroids are expressed with respect to the coordinate system of the agent, and, similar to the semantic maps, updated over time using egomotion. We track previously detected objects by their 3D centroid $C \in \mathbb{R}^3$. We estimate the centroid by taking the 3D point corresponding to the median depth within the segmentation mask and bring it to a common coordinate frame. We do a simple form of non-maximum suppression on the object memory, by comparing the euclidean distance of centroids in the memory to new detected centroids of the same category, and keep the one with the highest score if they fall within a distance threshold.

For each object in the object memory, we maintain an object state dictionary with a pre-defined list of attributes. These attributes include: category label, centroid location, holding, detection score, can use, sliced, toasted, clean, cooked. For the attributes, these are initialized by sending the detected object crops in the abstracted state, defined by the detector mask, to the VLM model, and asking it "Is this {category} {word1} or {word2}? Provide only your answer, either "{word1}" or "{word2}", and taking the answer as the output attribute.

### S6.2  VisualWebArena

The VisualWebArena [37] builds on Web Arena [92] contains 910 evaluation instructions with three interactive websites: Classifieds, Reddit, and Shopping. At each time step, the agent obtains a Set of Marks annotated image and and the webpage content, in a textual format listing the button text with their set of marks ID. The set of marks bounding boxes and textual state are extracted from the HTML code for the current webpage. At each time step, the agent must select an action to carry out the instruction. The instruction includes a natural language description and potentially one or more reference images. The action space is as follows:

- click [elem] Click on element elem.
- hover [elem] Hover on element elem.
- type [elem] [text] Type text on element elem.
- press [key comb] Press a key combination.
- new tab Open a new tab.
- tab focus [index] Focus on the i-th tab.
- tab close Close current tab.
- goto [url] Open url.
- go back Click the back button.

- go forward Click the forward button.
- scroll [up|down] Scroll up or down the page.
- stop [answer] End the task with an optional output

### S6.3 Additional details on ICAL agent deployment in VisualWebArena

At each time step, we retrieve the top-5 examples and prompt the model with the 5 in-context examples. Each in-context example consists of the image input, abstracted textual state, summary, step-by-step reasoning, predicted next state, abstraction comments, and predicted action. We use the Set of Marks (SoM) [84] representation for image inputs, implemented in VisualWebArena by [37]. An example in-context example is shown in Listing S4.

Listing S4: In-context example used in VisualWebArena. Note that the webpage screenshot with SoM annotations for the in-context example is also provided to the VLM, but is not displayed.

```
Input:

OBJECTIVE: I recall seeing this exact item of pillows in the Household section on the
    site, add a comment on its listing with the title "Commentary" and text "How funky
    ...".

OBSERVATION:
[4] [A] [Publish Ad]
[] [StaticText] [> Search results: pillows]
[8] [INPUT] []
[] [StaticText] [Min.]
[15] [A] [Household]
[18] [A] []
[] [StaticText] [Listings]
[] [StaticText] [North Potomac  (Maryland)]
[] [StaticText] [15.00 $]
[] [StaticText] [Pottery Barn Matine Drape (1 panel)                  20.00 $
                    Household              Lake Barcroft  (Virginia)
    2023/11/11                       Beaut]
[] [StaticText] [100.00 $]
[33] [IMG] [Comforter with Pillows, description: a green quilt and pillows on the ground,
     url: oc-content/uploads/1243/1243_thumbnail.png]
[] [StaticText] [Roanoke  (Virginia)]
[] [StaticText] [Comforter with 2 Shams and 1 Throw Pillow 7'1 x 7'3]

PREVIOUS ACTIONS:
1. type  [INPUT] [] [pillows]
2. hover  [SELECT] [Select a categoryAntiquesAppliancesArts + craftsAuto partsBeauty +
     healthBikesBoatsBooksCars + trucksCell phonesCollectiblesComputer
     partsComputersElectronicsFarm + gardenFurnitureHouseholdJewelryMoto]
3. click  [SELECT] [Select a categoryAntiquesAppliancesArts + craftsAuto partsBeauty +
     healthBikesBoatsBooksCars + trucksCell phonesCollectiblesComputer
     partsComputersElectronicsFarm + gardenFurnitureHouseholdJewelryMoto]
4. click  [A] [Household]
5. type  [INPUT] [] [pillows]
6. click  [BUTTON] [Apply]
7. scroll [down]
8. click [25] [A]
9. scroll [down]
10. click  [A] [2]
11. click  [A] [Household]
12. scroll [down]
13. click  [A] [2]

Output:

Plan: I need to find the exact item of pillows with the title "Commentary" to add a
    comment. I've already searched for pillows, filtered by the Household category, and
    navigated through the pages. The item I'm looking for is not visible on the current
    screen, so I must continue to scroll down to find it. Once I locate the item, I can
    proceed to add the comment with the specified text "How funky...".

Summary: Scrolling down will allow me to find the exact pillows listing where I need to
    add the comment, progressing towards the task completion.

Predicted Next State: The next state will display additional listings of household items,
    including more pillow options, as the page is scrolled down to reveal further
    content that was previously not visible.

Action: In summary, the next action I will perform is ```scroll [down]```

Abstraction comments:
1. When the end of a page is reached and the task requires viewing more items, navigate
    to the next page instead of attempting to scroll further.
2. When posting a comment on a classifieds website, ensure that all required fields, such
    as the email address, are filled in before submitting the comment.
3. When searching for items in a specific location on a classifieds website, it is
    important to use the location filter to narrow down the search results to the
    desired area before applying the filter.
4. When a comment has been typed into the appropriate field, the next step is to submit
    the comment using the provided submission button, rather than retyping or editing
    the comment.
5. When searching for items within a specific price range, it is essential to set the
    minimum and maximum price filters before applying the search to narrow down the
    results.
6. When tasked with selecting a specific item in a sequence, ensure the correct order is
    followed based on the given instructions. In this case, the item must be selected (
    clicked on) before completing the task.
```

```
7. When the objective is to leave a comment with both a title and text, ensure that both
   fields are completed before submitting the comment.
```

## S6.4    Ego4D

Ego4D is a daily life activity video dataset of hundreds of scenarios. We focus on the long-term action anticipation task to predict the future user actions given an RGB egocentric video. Models must choose from 115 verbs and a set of 478 nouns for action predictions. For evaluation, we take 100 seen validation videos that come from the same videos used for ICAL example learning but at a different, unseen location, and a separate 200 completely unseen validation videos for evaluation. We follow previous work and use edit distance as a metric, which is computed as the Damerau-Levenshtein distance over sequences of predictions of verbs, nouns and actions. The goal of this measure is to assess performance in a way which is robust to some error in the predicted order of future actions. All GPT4V evaluations give image inputs annotated with DEVA tracking masks [12] with Set-of-Marks labels [84]. For in-context examples to GPT4V, we concatenate 3 uniformly spaced video frames and give it as a single image input. For the input video to GPT4V, we take 12 video frames uniformly spaced and provide 4 images each with 3 concatenated frames. The supervised baseline uses a SlowFast backbone with a Transformer aggregator and trains on Ego4D V2 (243 video hrs) [28].

### S6.4.1    Noisy Trajectories

100 demonstrations from validation set were abstracted using ICAL. Due to the passive nature of this task, we perform ICAL without the abstraction verification with a human-in-the-loop phase, and only perform the VLM-driven Abstraction Generation (Section 3.2). 92 demonstrations (8 failed due to GPT4V filters) were successfully abstracted by ICAL for an example set size of 92 for evaluation.

## Listing S5: Prompt template for VLM abstraction generation phase

```
**Objective:** As a helpful assistant with expertise in {DOMAIN}, your task is to produce
    useful abstractions and language comments to help someone else perform the task.

**Information Provided:**
You will receive:
{INPUT INFORMATION}

**Output Format:**
1. Summary: Provide a summary of the task the user is performing. Start this with '
    Summary:" and limit it to a single line, no more than 6 sentences.
2. Abstracted State: List the elements that are relevant for the task that the user is
    performing and are important for the task. Refer to the elements by their object ID,
    and for each element, a description of the object and and relevant attributes.
    Start the list with 'Abstracted State:', and put each element that you choose on a
    new line.
3. Step-by-step Reasoning: Explain each step of the demonstration and the reasoning for
    each step. Mention specific object numerical IDs when referencing objects. Start
    this section with "Step-by-step Reasoning:" and limit it to a single line, no more
    than 6 sentences.
4. Predicted State Change: Provide in natural language any relevant state changes of
    objects and visual elements that will take place due to future actions. Remember to
    focus on state changes that will help someone else perform the task.
5. Abstraction Comments: Provide a numbered list of useful language abstraction comments,
    such as causal abstractions, task abstractions, and other abstractions that will
    help someone learn the task. Put each abstraction on a new line. Mention specific
    object IDs when referencing objects.
6. Optimized Demonstration Script: Present any optimized actions for completing the task
    more efficiently in the current environment. It is possible that the provided
    demonstration script is already optimally efficient and no revisions are needed.

**Action Space**
{ACTION API}

**In-Context Examples:**
{EXAMPLES}

**Guidelines:**
Follow these strict guidelines:
1. Adhere to the previously defined output format without deviating. Refer to the
    examples provided for proper format.
2. Reason through each step methodically, as shown in examples.
3. Reference object/part IDs in your reasoning when it's relevant.
4. Your primary focus should be on generating useful comments that will help someone else
    accurately perform the task.
```

## Listing S6: Prompt template for human-in-the-loop revisions based on human feedback

```
**Objective:** You are an autonomous intelligent agent tasked with {DOMAIN}. Your primary
    goal is to revise an action taken on a website based on natural language corrective
    feedback so that the action successfully makes progress towards completing the task
    .

**Information Provided:**
Here's the information you'll have:
{INPUT INFORMATION}

**Output Format:**
1. Explain: Why does the action not complete the task? What does the human feedback imply
    ? What revisions should be made to fix the error? This should be a single line, and
    at most six sentences.
2. Summary: Single-line summary of what the proposed new action will carry out and how it
    will make progress towards the objective.
3. Abstracted State: List the elements that are relevant for the task that the user is
    performing and are important for the task. Refer to the elements by their object ID,
    and for each element, a description of the object and and relevant attributes.
4. Step-by-step Reasoning: Explain each step of the demonstration, the reasoning for each
    step, and why the revised action would make the most sense.
5. Predicted State Change: Predict what the next state will look like after taking the
    proposed revised action.
6. Correction Abstraction:
Provide a numbered list of useful language abstraction comments, such as causal
    abstractions, task abstractions, and other abstractions that will help someone learn
    the task. Put each abstraction on a new line. Mention specific object IDs when
    referencing objects. Also, incorporate the correction into some generalizable
    knowledge about the error, why it is a mistake, and how to fix it.
7. Revised Action: Output the revised action to take from the actions provided below.

**Action Space**
{ACTION API}

**In-Context Examples:**
{EXAMPLES}

**Guidelines:**
Follow these strict guidelines:
1. Adhere to the previously defined output format without deviating. Refer to the
    examples provided for proper format.
2. Reason through each step methodically, as shown in examples.
3. Reference object/part IDs in your reasoning when it's relevant.
4. Your primary focus should be on generating useful comments that will help someone else
    accurately perform the task.
```

Listing S7: Prompt template for VLM planning after examples are learned

```
**Objective:** As a helpful assistant with expertise in {DOMAIN}, your task is to {DOMAIN
    TASK}

**Information Provided:**
You will receive:
{INPUT INFORMATION}

**Output Format:**
1. Summary: Provide a summary of the task you are performing. Start this with 'Summary:"
    and limit it to a single line, no more than 6 sentences.
2. Abstracted State: List relevant objects in the scene by their numerical IDs, providing
    a description and any pertinent attributes for each. Start the list with '
    Abstracted State:', and put each element that you choose on a new line.
3. Step-by-step Reasoning: Explain each step of the task and the reasoning for each step.
    Mention specific object numerical IDs when referencing objects. Start this section
    with "Step-by-step Reasoning:" and limit it to a single line.
4. Predicted State Change: Provide in natural language any relevant state changes that
    will occur throughout the task.
5. Abstraction Comments: Provide a numbered list of useful language abstraction comments,
    such as causal abstractions, task abstractions, and other abstractions that will
    help someone learn to predict the future actions from the egocentric video. Put each
    abstraction on a new line. Mention specific object numerical IDs when referencing
    objects.
6. Predicted Actions: Present the actions the agent should take to carry out the task.

**Action Space:**
{ACTION API}

**In-Context Examples:**
{RETRIEVED EXAMPLES}

**Guidelines:**
Follow these strict guidelines:
1. Adhere to the previously defined output format without deviating. Refer to the
    examples provided for proper format.
2. Reason through each step methodically, as shown in examples.
3. Reference object/part IDs in your reasoning when it's relevant.
```

