# OpenReview forum: "VLM Agents Generate Their Own Memories: Distilling Experience into Embodied Programs of Thought"
_NeurIPS.cc/2024/Conference — NeurIPS 2024 spotlight_

### Official Review · Reviewer_foCb · 2024-07-11

**Soundness:** 3
**Presentation:** 2
**Contribution:** 2
**Rating:** 5
**Confidence:** 4

**Summary:**

This work proposed ICAL, aims to improve decision-making in large language and vision-language models by generating optimized trajectories and language annotations from noisy demonstrations and human feedback. ICAL abstracts noisy trajectories into optimized sequences with language comments, refined through human feedback during execution. The experiment shows ICAL significantly improves performance in benchmarks like TEACh, VisualWebArena, and Ego4D, surpassing state-of-the-art methods.

**Strengths:**

1、Improved Performance: ICAL significantly enhances decision-making and task success rates across various benchmarks, such as TEACh, VisualWebArena, and Ego4D.

2、Reduced Reliance on Expert Examples: The method minimizes the need for expert-crafted examples by generating useful abstractions from sub-optimal demonstrations and human feedback.

3、Versatility: ICAL is effective in multiple domains, including dialogue-based instruction following, multimodal web tasks, and video action anticipation.

4、Human Feedback Integration: The approach incorporates human feedback to refine and adapt abstractions, improving the agent’s performance over time.

**Weaknesses:**

1、The method still relies on human feedback for refining abstractions, which may not always be feasible or scalable.

2、The process of generating and refining abstractions heavily rely on GPT4-V

3、The effectiveness of ICAL is constrained by the capabilities of the underlying Vision-Language Models (VLMs), such as GPT-4V.

4、The method’s performance can be affected by the quality of the initial noisy demonstrations and the accuracy of human feedback.

**Questions:**

The VLM-driven Abstraction Generation component appears to heavily rely on the performance of GPT-4V. There is no ablation study for replacing GPT-4V. If such results exist and I missed them, please indicate where they can be found.

**Limitations:**

Yes

---

> ### Author Rebuttal · Authors · 2024-08-06
>
> >**ICAL's effectiveness is constrained by the capabilities of VLMs like GPT-4V. The VLM-driven Abstraction Generation relies heavily on GPT-4V. Is there an ablation study for its replacement?**
>
> We appreciate the comment regarding the reliance on GPT-4V for the VLM-driven Abstraction Generation component. We acknowledge the importance of evaluating alternative models and have addressed this in our work.
>
> In Table S2 of the Appendix, we provided an ablation study using GPT-3.5 for the VLM-driven Abstraction Generation on the TEACh dataset, where visual experiences are converted to textual format. The results indicate a performance drop when using GPT-3.5 for ICAL Abstraction Generation, with the number of successfully revised trajectories reduced by more than half compared to GPT-4V (52 versus 122 tasks successfully completed). In Section S3.4 of the Appendix, we demonstrate how relabeling unsuccessfully refined trajectories can help bridge this performance gap when using weaker models like GPT-3.5, thereby reducing the dependence on GPT-4V. We will highlight these findings more prominently in the main paper to ensure their visibility. At the time of our experiments, no open-source VLMs with in-context multimodal generation capabilities comparable to GPT-4V were available. Per your comments, we implemented LLaVa-NeXT for the abstraction phase in VisualWebArena given multimodal inputs. As shown in Figure R2, we found that it failed to properly revise the abstractions to take into account the feedback. However, we are committed to continuing our experiments with emerging open-source VLMs as they become available.
>
> >**The method still relies on human feedback for refining abstractions, which may not always be feasible or scalable.**
>
> It is essential to note that this feedback is sparse, provided in natural language, and required only a few times for each task. In the Visual Web Arena, our agent needs an average of just 5.36 natural language feedbacks per example, with each feedback taking less than 15 seconds to create. In TEACh, our agent needs just 1.516 average feedbacks per episode with an average length of 18 words per natural language feedback. In fact, natural language feedback in ICAL could be easily communicated via speech by a human while observing the agent, whereas low-level coding and abstraction writing communication would be significantly more laborious. In order to hand-write GPT4-V-length abstractions, this would require 202.62 words on average for each example, including the precise coding of actions.
>
> **Additionally, our agent becomes increasingly efficient over time, requiring less human feedback and fewer environment interactions as it processes more examples.** By retrieving past successful abstractions during the VLM-abstraction making and human-in-the-loop phases, it uses previously stored knowledge to help abstract new examples. **As shown in Figure R1, for the second half of examples processed, the model requires significantly fewer environment steps (436±88 vs. 267±43, p=0.0143) and human feedbacks (0.74±0.17 vs. 0.21±0.08, p=0.0089) per episode. This demonstrates that retrieving abstracted examples during abstraction learning reduces both human effort and environment interaction over time.** Consequently, using previously stored ICAL examples not only improves test performance but also accelerates learning for future examples.
>
> Furthermore, the feedback provided does not require specialized expertise. For example, typical feedback from VisualWebArena includes comments like, "This does not have 2 upvotes and includes a meme of three Spider-Men, a flag of the Netherlands, and a flag of Croatia. You should scroll down to see more posts." In TEACh, feedback might be, "The sink is full right now. Empty the sink before interacting with it." Such straightforward feedback significantly reduces the complexity and cost of data collection. It also often provides more context for the model to create generalizable abstractions. For example, the sink being full is not directly necessary for correcting the mistake, but will help the agent learn generalizable knowledge via the generated abstractions.
>
> Our approach additionally minimizes the need for detailed annotations or extensive interactions required to train reinforcement learning (RL) or behavior cloned agents for long-horizon, multimodal tasks. This method not only shortens the data collection process but also ensures minimal human expertise and interaction, making it a scalable and cost-effective alternative.
>
> >**The method’s performance can be affected by the quality of the initial noisy demonstrations and the accuracy of human feedback.**
>
> We acknowledge the concern regarding the impact of the quality of initial noisy demonstrations and the accuracy of human feedback on our method. Our method is designed to handle and successfully revise even very noisy or incorrect demonstrations.
>
> For instance, Listings S2 and S3 in the Appendix of the paper illustrates a very noisy code trajectory that was successfully corrected by ICAL. While it is true that ICAL may not always recover from extremely noisy demonstrations or feedback, such cases are typically filtered out or relabeled, as discussed in step 6 of Section 3.3 in the main paper.
>
> This challenge is not unique to our method; gradient-based methods also suffer from inaccurate examples that can lead to incorrect model updates. As suggested by reviewer vg3K, one viable solution to this could be to utilize a reward model to identify and remove misleading or extremely noisy demonstrations before processing.

---

> > ### Comment · Reviewer_foCb · 2024-08-13
> > **increase rate to 5**
> >
> > Thanks to the authors for the detailed response. My concern about the dependency on human feedback has been addressed. So I increase my rate from 4 to 5. I do not give a higher rate at the current stage. That is because I am still concerned about the generalization and reproducibility of the overall flow, as it heavily relies on the close-sourced GPT-4. If the proposed method can not help any open-source models, it is hard to justify that the contribution can be generalized to other VLMs.

---

> ### Author Response · Authors · 2024-08-13
> **Thank you!**
>
> Thank you for raising your score from 4 to 5. We truly appreciate your feedback and will incorporate the discussion into the final version.
>
> We will include further discussion on our use of close-sourced VLMs in the paper. We are committed to continuing our experiments with emerging open-source VLMs as they become available. Additionally, we will emphasize our ablation studies with GPT-3.5 and fine-tuning in the main paper to address these points.

---

### Official Review · Reviewer_vg3K · 2024-07-11

**Soundness:** 3
**Presentation:** 3
**Contribution:** 3
**Rating:** 6
**Confidence:** 4

**Summary:**

The paper proposes a pipeline for Large Language and Vision-language models (LLMs and VLMs) to digest and learn from sub-optimal demonstrations and human feedback. The LLM/VLM, given sub-optimal task demonstrations, is prompted to produce abstractions of the trajectory (including task and causal abstractions, state changes, task decomposition and subgoals, and state abstractions), and possibly refine the given trajectory with insight from these abstractions. Optimally, these produced abstractions can be further refined by interactions with humans. Experiments show that the proposed framework greatly surpasses zero-shot CoT baselines and is competitive with/slightly better than having expert demonstations.

**Strengths:**

- The authors provide sufficient details with their experiment environments and experiment setup, as well as the code, for readers to better understand and reproduce the results.
- The paper proposes a novel method for LLMs to autonomously refine sub-optimal trajectories and build a high-level abstract understanding of the task based on sub-optimal trajectories. For very complex tasks, these high-level abstract descriptions have the potential to help humans better understand the task and the general strategies to accomplish the task.
- The experiments span three different types of environments, demonstrating the generality of the proposed method. The proposed method also achieve good experimental results over the baseline methods.

**Weaknesses:**

- The experiment result focuses on the accuracy of the method; not enough comparison is provided in terms of the efficiency. For example, whether is might be more efficient for the human feedback provider to directly edit and improve the sub-optimal trajectories and provide the human-refined ones to the LLM/VLM?
- The scaling capability of the proposed method is unclear. In the proposed method, each trajectory needs to go through the human-in-the-loop fine-tuning process which seems quite inefficient. For simple tasks in the TEACh benchmark the method needs ~100 trajectories to perform well according to Figure 5. For more complex tasks it might be less sample efficient and difficult to scale.
- It also requires the environment to automatically reset itself, which is another limitation.
- Some text in figures 1 and 2 are too small and difficult to read and understand. A more illustrative example of what the task is, what are the inputs/outputs, and what kind of information is included in the abstracted state is desired.

**Questions:**

- Can we easily design a reward function that specifies how sub-optimal a trajectory is? If we provide these sub-optimal trajectories and their corresponding rewards to the LLM/VLM, can it use this reward information to further refine the trajectories? The "in-context RL" capability of LLMs are previous studied in works like [1].
- What is the difference between expert and non-expert feedback (mentioned in lines 281-282)? Examples with qualitative differences can be very informative.

[1] Exploring Large Language Models for Communication Games: An Empirical Study on Werewolf. https://arxiv.org/abs/2309.04658

**Limitations:**

The authors fail to include sufficient details on the human-in-the-loop fine-tuning phase. The evaluation might be biased if the human feedback provider is very familiar with the task and the LLM; they might be able to provide much more effective feedback than non-experts. If this is the case, then the amount of training required for the human to provide more effective feedback is worth investigating.

---

> ### Author Rebuttal · Authors · 2024-08-06
>
> >**Efficiency of ICAL method?  For simple tasks in the TEACh benchmark the method needs ~100 trajectories to perform well according to Figure 5. Scaling capability of the human-in-the-loop phase?**
>
> Actually, these roughly 100 trajectories are used for all tasks in TEACh combined. In fact, each task has on average 9.67 tractories in the memory after ICAL learning. The TEACh dataset includes 12 complex, long-horizon task types, averaging 67 steps for the shortest task (water plant) and 359 for the longest task (breakfast) for a human expert to complete. This complexity is significantly greater than other popular benchmarks like ALFRED, which averages only 50 steps across all tasks for experts. Despite the complexity, ICAL only requires a few *noisy* trajectories for each task. This is 6x less in-domain data than the strongest behavior cloning baseline (E.T. [1]), and achieves 21X the success rate, while also using noisy demonstrations. In VisualWebArena and Ego4D, which also represent high complexity, the number of demonstrations used for ICAL is of a similar magnitude (~100).
>
> **Importantly, our agent becomes increasingly efficient over time, requiring less human feedback and fewer environment interactions as it processes more examples.** By retrieving past successful abstractions during the VLM-abstraction making and human-in-the-loop phases, it uses previously stored knowledge to help abstract new examples. **As shown in Figure R1, for the second half of examples processed, the model requires significantly fewer environment steps (436±88 vs. 267±43, p=0.0143) and human feedbacks per episode (0.74±0.17 vs. 0.21±0.08, p=0.0089). This demonstrates that retrieving abstracted examples during abstraction learning reduces both human effort and environment interaction over time.** Consequently, using previously stored ICAL examples not only improves test performance but also accelerates learning for future examples.
>
> It is essential to note that the human feedback is sparse, provided in natural language, and required only a few times for each task. In VisualWebArena, our agent needs an average of just 5.36 natural language feedbacks per example, with each feedback taking less than 15 seconds to create. In TEACh, our agent needs just 1.516 average feedbacks per episode with an average length of 18 words per natural language feedback. In fact, natural language feedback in ICAL could be easily communicated via speech by a human while observing the agent, whereas low-level coding and abstraction writing communication would be significantly more laborious. In order to hand-write GPT4-V-length abstractions, this would require 202.62 words on average for each example in VisualWebArena and 48-107 lines of text for each example in TEACh, including the precise programming of actions or code.
>
> >**Can we easily design a reward function that specifies how sub-optimal a trajectory is? Can the LLM/VLM use this reward information to further refine the trajectories?**
>
> Thank you for this interesting suggestion. Based on your comment, we prompted GPT-4 with the sub-optimal trajectory and retrieved examples from memory, and asked the model to assign an "optimality" score from 1 to 5 with a reflection explaining the score. This score and reflection were then given to the VLM during the abstraction generation phase. After running this experiment for 40,000 steps, we observed no significant difference in the number of tasks successfully completed (22 with the score and 24 without the score).
>
> We hypothesize that this is because the VLM already performs this evaluation implicitly. This is supported by Figure R1, where in-context examples reduce the need for human feedback and environment interaction over time. This demonstrates that the VLM effectively utilizes the provided examples during the learning phases to revise the trajectory, inferring sub-optimality without explicit scoring.
>
> >**What is the difference between expert and non-expert feedback? More details on the human-in-the-loop fine-tuning phase.**
>
> The authors and their lab colleagues provided the human-in-the-loop feedback in the paper. We did not recruit participants due to the costs associated with recruiting participants for running ICAL and ablations. However, the authors have no more familiarity with the websites used in VisualWebArena than a typical web user and did not provide expert feedback, meaning it did not focus on precise coding or low-level changes. We will make these details more clear in the main paper. For example, typical feedback from VisualWebArena includes comments like, "This does not have 2 upvotes and includes a meme of three Spider-Men, a flag of the Netherlands, and a flag of Croatia. You should scroll down to see more posts." In TEACh, typical feedback is of the form, "The sink is full right now. Empty the sink before interacting with it."
>
> >**It also requires the environment to automatically reset itself.**
>
> ICAL improves learning efficiency as more examples are learned, reducing human effort *and number of environment resets* over time (Figure R1). Our approach also minimizes the need for extensive resets required to train reinforcement learning (RL) or large data collection for behavior cloning. Despite this, some resetting is necessary and we will add this to the limitations of the method.
>
> >**Some text in figures 1 and 2 is too small and difficult to read and understand. A more illustrative example of the task, inputs/outputs, and abstracted state is desired.**
>
> Thank you for the feedback. We will increase the font size in Figures 1 and 2. Additionally, we will include an extra figure to clearly illustrate the task, inputs, outputs, and the abstracted state. This figure will specifically show the exact inputs (e.g., images, text) for VisualWebArena and the corresponding outputs from the abstraction generation phase.
>
> **References**
>
> [1] Pashevich et. al. (2021). Episodic Transformer for Vision-and-Language Navigation.

---

> > ### Comment · Reviewer_vg3K · 2024-08-08
> >
> > I thank the authors for addressing my questions. Considering the overall contributions and limitations of the paper, I will keep my ratings.

---

> > > ### Author Response · Authors · 2024-08-13
> > > **Thank you!**
> > >
> > > Thank you for your thoughtful feedback. We appreciate your recognition of the strengths of our work and your constructive suggestions. We will include the clarifications and feedback in the final version. If you have any further questions, please feel free to discuss them with us.
> > >
> > > We are particularly pleased that the suggested experiments and feedback allowed us to emphasize the human-in-the-loop efficiency and further clarify the human feedback and trajectory refinement.

---

### Official Review · Reviewer_ih5w · 2024-07-12

**Soundness:** 3
**Presentation:** 3
**Contribution:** 2
**Rating:** 5
**Confidence:** 4

**Summary:**

The paper proposes ICAL, In-Context Abstraction Learning, which builds a memory of suboptimal experiences that are abstracted into states and plans, as well as correction and reflection from human feedback. The approach is based on extensive prompting to elicit structured representations from past experiences, and apply RAG at test time. ICAL exhibits improvement in success rates across virtual domains compared to raw VLMs and CoT methods.

**Strengths:**

The idea of using extensive prompting to generate structured representation for RAG is clean and intuitive, and the authors clearly detail the steps involved. Using RAG enables the overall pipeline to improve continually.

**Weaknesses:**

My concern with the paper is the limited contribution in terms of the approach. It is a relatively simple adaptation of previous methods such as CoT, ReACT, Socratic Model [1] as well as RAG.

[1] Zeng, Andy, et al. "Socratic models: Composing zero-shot multimodal reasoning with language." arXiv preprint arXiv:2204.00598 (2022).

**Questions:**

Could you further discuss the limitations of the work? I think currently the paper lacks thorough discussions on the limitations. For example, what are the common failure modes? And is there a viable and scalable solution to it potentially?

**Limitations:**

The overall prompting strategy is quite sensitive to the model performance. The fine-tuning improvement is quite limited as discussed in 4.6.

---

> ### Author Rebuttal · Authors · 2024-08-06
>
> >**Clarifying the novel contributions and advancements beyond existing methods, such as CoT, ReACT, Socratic Model, or RAG.**
>
> While our framework incorporates chain of thought prompting from CoT, interleaves reasoning and acting as seen in ReACT, predicts plans over pretrained modules similar to Socratic models, and uses retrieval-augmented generation of action plans inspired by RAG methods, it is distinct in its primary contribution: optimizing examples for improved in-context learning. We leverage VLMs to create abstractions for examples, improving their utility as RAG inputs. To the best of our knowledge, this is the first work to do this. Specifically, we introduce a novel two-stage approach that refines each example and generates four types of generalizable knowledge, enabling rapid adaptation of multimodal agents with few demonstrations. Our results, in Tables 1-3, demonstrate that both stages are crucial for high performance, significantly outperforming CoT and RAG, which use raw or handwritten in-context examples. Additionally, Figure 5 illustrates our method's support for continual learning without forgetting.
>
> Furthermore, our optimized in-context examples are applicable to various multimodal agents across three complex domains: long-horizon instruction-following for household robots, visually-grounded web navigation, and action forecasting from real-world egocentric videos. Reviewers vg3K and VRBk note that these experiments on "sufficiently different tasks... suggest the generalizability of the proposed method." Previous works have been limited to text-based environments, gaming scenarios, or single real-world domains. To our knowledge, we are the first to show robust results across these three diverse and complex domains.
>
> We position our approach as a superset that incorporates the strengths of previous VLM agent works, while introducing a novel focus on optimizing in-context learning examples. As Reviewer VRBk puts it, ''the overall goal of using VLMs to distill demonstrations into canonical examples is an interesting combination of foundation models with classical ideas such as case-based learning.'' Our approach can be likened to in-context policy improvement with a multi-modal LLM actor, where instead of aiming to maximize rewards through specific reasoning strategies explored in previous works, it refines and optimizes in-context examples.
>
> >**The overall prompting strategy is quite sensitive to the model performance. The fine-tuning improvement is quite limited as discussed in 4.6.**
>
> We apologize for not clarifying these points in our paper. Specifically, we show that fine-tuning CoT-prompted GPT-3.5 significantly improves performance, doubling its success rate from 11.8% to 23.2%. Additionally, incorporating retrieval-augmented generation (RAG) with fine-tuning (using ICAL examples as memory) provides further improvement, resulting in our best-performing agent. This agent outperforms the non-finetuned memory-augmented CoT-prompted GPT-3.5 by 4.9% in goal-condition success.
>
> We view weights and external memory of abstractions as two forms of memory with distinct benefits. Weight fine-tuning requires many examples, while RAG can learn from a single example. In scenarios with limited data (150 examples or fewer), external memory updates and RAG use data more efficiently and remain competitive with weight fine-tuning. This is relevant to our study as our considered domains fit this category.
>
> >**Discussion on the limitations of our work.**
>
> We will expand our limitation section per your request. Below, we include additional limitations and error modes of our agent with specific examples. We will include this discussion in the main paper.
>
> 1. **Visual Grounding Limitations of GPT-4V**
> While ICAL improves performance in visually grounded tasks, errors persist due to the base VLM's limitations in visual grounding. For example, the agent fails to identify colors accurately, leading to errors like selecting the wrong item. This issue is evident in the "reddit_2" and "reddit_9" web tasks, where the agent navigates to incorrect posts, showing a failure to match images to webpage tabs. The agent also occasionally struggles with pre- and post-condition recognition. We found that in-context multimodal examples helped grounding in cases where grounding elements were unique, but easily identifiable but continue to fail in fine-grained cases. This grounding limitation has been noted in previous work as well [1]. We expect improvements with more data, added grounding objectives [2], and fine-grained image annotations [3]. In future work, we plan to extend ICAL for better fine-grain language grounding using multimodal retrieval, building on methods like ViperGPT [4].
>
> 2. **Fine-grained in-context planning failures**
> During learning phases, ICAL may not fully acquire all necessary information for test time, resulting in failures. For instance, in the "shopping_3" web task, where the instruction is to display the most expensive red controller from the "PS4 accessories" category, the agent navigates to the PS4 category but fails to access the accessories subcategory. This demonstrates a lack of understanding of website structures and navigation. This limitation suggests that agents need to recognize when their knowledge base is insufficient and query the user for missing information. We plan to address active querying during testing in future work.
>
> **References**
>
> [1] Zheng et. al. GPT-4V(ision) is a Generalist Web Agent, if Grounded.
>
> [2] Ma et al. (2024). Groma: Localized Visual Tokenization. arXiv:2404.13013.
>
> [3] Garg et al. (2024). ImageInWords: Hyper-Detailed Image Descriptions. arXiv:2405.02793.
>
> [4] Suris et. al. (2023) ViperGPT: Visual Inference via Python Execution for Reasoning.

---

> ### Comment · Reviewer_ih5w · 2024-08-13
> **Response to the rebuttal**
>
> I thank the authors for answering my questions and addressing my concerns. I am more convinced with the contribution of optimizing few-shot examples now and how it differs from previous work. I will raise my score to 5.
>
> I am still generally concerned with the overall limitation of the work ---  I am glad the authors provide additional discussions on it and I hope they will be added in the revised manuscript, but I have to say I am generally less convinced of prompting techniques that build on previous strategies, despite the performance improvement that the authors demonstrate.

---

> > ### Author Response · Authors · 2024-08-13
> > **Thank you!**
> >
> > Thank you for your thoughtful feedback and for raising your score to 5. We appreciate your recognition of our work in optimizing few-shot examples and clarifying how our approach differs from prior work.
> >
> > We understand your concerns about building on existing strategies. In the revised manuscript, we will include a thorough discussion of the limitations and how our approach differs from previous work.

---

### Official Review · Reviewer_VRBk · 2024-07-12

**Soundness:** 4
**Presentation:** 4
**Contribution:** 4
**Rating:** 9
**Confidence:** 4

**Summary:**

The goal of this paper is to teach VLMs novel tasks by prompting VLMs to create multimodal abstractions for unfamiliar domains. Given instructions paired with noisy demonstration trajectories, this paper proposes a method to encapsulate the information from these into examples consisting of optimized trajectories paired with generalizable language abstractions. A VLM is specifically prompted to produce abstractions such as essential task steps, cause and effect generalizations, expected state changes, a step by step plan, and relevant parts of state information. The abstracted example includes an executable trajectory which is executed and potentially refined with a human in the loop via natural language feedback. These examples are then used for in context of a VLM to improve task performance.
The paper includes experiments on 3 datasets - TEACh, VisualWebArena and Ego4D, comparing respectively to the HELPER model - a prior SoTA model for TEACh, and GPT4V for VisualWebArena and Ego4D. The most improvement in seen in VisualWebArena, followed by partial success on TEACh. On Ego4D, performance is on par with supervised learning methods that use more data but Zero shot CoT with GPT4V performs better.

**Strengths:**

The overall goal in this paper of using VLMs to distill demonstrations into canonical examples is an interesting combination of currently popular foundation models with more classical ideas such as case based learning.
The paper includes experiments on multiple benchmarks with sufficiently different tasks to suggest generalizability of the proposed method. It is also refreshing to see that the authors do not hesitate to include the unusually good performance of Zero shot GPT4V on Ego4D which might be seen as a negative result.
Given the topic of the paper, a number of engineering details and information about prior work is needed to fully understand the paper and increase the likelihood of reproducibility of the work. The authors have made a very strong attempt at this with a detailed appendix.

**Weaknesses:**

While the appendix goes a long way towards reducing this, it is likely that the paper might still be difficult to follow for readers unfamiliar with the datasets and prior methods referenced in this paper.

**Questions:**

It is possible I missed this in the paper, but I how was the human in the loop feedback obtained for the TEACh experiment? Given that the checklist says no human subjects were recruited, I assume this feedback was provided by authors. Is this correct?

**Limitations:**

The authors include a discussion of limitations and potential negative social impact.

---

> ### Author Rebuttal · Authors · 2024-08-06
>
> >**While the appendix goes a long way towards reducing this, it is likely that the paper might still be difficult to follow for readers unfamiliar with the datasets and prior methods referenced in this paper.**
>
> Thank you for your feedback. We acknowledge that the paper may still be challenging for readers unfamiliar with the datasets and prior methods. To address this, we will provide more detailed explanations on dataset processing and implementation in the appendix in the camera-ready version and the README in the code repository.
>
> >**It is possible I missed this in the paper, but how was the human in the loop feedback obtained for the TEACh experiment? Given that the checklist says no human subjects were recruited, I assume this feedback was provided by authors. Is this correct?**
>
> Yes, the authors and their lab colleagues provided the human-in-the-loop feedback in the paper. We did not recruit participants due to resource constraints, specifically the costs associated with recruiting participants for running ICAL and ablations. During this phase, we translated task progress into natural language feedback during failures, including failed actions (e.g., “The toaster is full right now.”) and missed task steps (e.g., “The pillow needs to be put onto the sofa”). The feedback provided did not require domain expertise, such as focusing on precise coding or low-level action changes. For example, typical feedback from VisualWebArena includes comments like, "This does not have 2 upvotes and includes a meme of three Spider-Men, a flag of the Netherlands, and a flag of Croatia. You should scroll down to see more posts." In TEACh, typical feedback is of the form, "The sink is full right now. Empty the sink before interacting with it." We will make these details more clear in the main paper.
>
> >**Performance comparison on Ego4D.**
>
> To clarify, zero-shot CoT with GPT-4V shows strong results, but ICAL with GPT-4V outperforms it. Specifically, ICAL achieves lower edit distances relative to GPT-4V zero-shot COT: increases in performance by 9.9 for verbs, 10.0 for nouns, and 4.0 for actions.

---

### Author Rebuttal · Authors · 2024-08-06

We appreciate the reviewers' positive feedback. Reviewer VRBk commended our innovative combination of foundation models with case-based learning, robust generalizability across benchmarks, and transparency in results. Reviewer ih5w highlighted ICAL's improved success rates across domains, and our clean, intuitive structured representation. Reviewer vg3K noted our framework's advancements over zero-shot CoT baselines, detailed experimental setup aiding reproducibility, and potential for helping humans understand complex tasks. Reviewer foCb noted ICAL's superior performance across various domains and continual integration of human feedback to improve over time.

We address overall concerns in this global response and specific reviewer concerns in their respective responses.

>@Reviewers vg3K, foCb: **How efficient is the human-in-the-loop phase? Is it scalable?**

**Our agent becomes increasingly efficient over time, requiring less human feedback and fewer environment interactions as it processes more examples.** By retrieving past successful abstractions during the VLM-abstraction making and human-in-the-loop phases, it uses previously stored knowledge to help abstract new examples. **As shown in Figure R1, for the second half of examples processed, the model requires significantly fewer environment steps (436±88 vs. 267±43, p=0.0143) and human feedbacks (0.74±0.17 vs. 0.21±0.08, p=0.0089) per example. This demonstrates that retrieving abstracted examples during abstraction learning reduces both human effort and environment interaction over time.** Consequently, using previously stored ICAL examples not only improves test performance but also accelerates learning for future examples.

It is additionally essential to note that this feedback is sparse, provided in natural language, and required only a few times for each task. In VisualWebArena, our agent needs an average of just 5.36 natural language feedbacks per example, with each feedback taking less than 15 seconds to create. In TEACh, our agent needs just 1.516 average feedbacks per episode with an average length of 18 words per natural language feedback. In fact, natural language feedback in ICAL could be easily communicated via speech by a human while observing the agent, whereas low-level coding and abstraction writing communication would be significantly more laborious. In order to hand-write GPT4-V-length abstractions, this would require 202.62 words on average for each example in VisualWebArena and 48-107 lines of text for each example in TEACh, including the precise programming of actions or code.

>@Reviewers foCb, ih5w: **Does the method rely on GPT4V? How sensitive is it to the prompting strategy?**

In Table S2 of the Appendix, we provided an ablation study using GPT-3.5 for the VLM-driven Abstraction Generation on the TEACh dataset, where visual experiences are converted to textual format. The results indicate a performance drop when using GPT-3.5 for ICAL Abstraction Generation compared to GPT4V (52 versus 122 tasks successfully completed). In Section S3.4 of the Appendix, **we demonstrate how relabeling unsuccessfully refined trajectories can help bridge this performance gap when using weaker models like GPT-3.5, thereby reducing the dependence on GPT-4V.** We will highlight these findings more prominently in the main paper to ensure their visibility.

Per your comments, we implemented LLaVa-NeXT for the abstraction phase in VisualWebArena given multimodal inputs. As shown in Figure R2, we found that it failed to properly revise the abstractions to take into account the feedback. However, we are committed to continuing our experiments with emerging open-source VLMs as they become available.

We further show in Section 4.6 that fine-tuning CoT-prompted GPT-3.5 significantly improves performance, doubling its success rate from 11.8% to 23.2%. Additionally, incorporating retrieval-augmented generation (RAG) with fine-tuning (using ICAL examples as memory) provides further improvement, resulting in our best-performing agent. This agent outperforms the non-finetuned memory-augmented CoT-prompted GPT-3.5 by 4.9% in goal-condition success. In scenarios with limited data (150 examples or fewer), external memory updates and RAG use data more efficiently and remain competitive with weight fine-tuning.

>@Reviewers ih5w: **How does this differ from CoT, ReACT, Socratic Model, and RAG?**

While our framework incorporates CoT, interleaves reasoning and acting as seen in ReACT, predicts plans over pretrained modules similar to Socratic models, and uses RAG for inference, it is distinct in its primary contribution: optimizing examples for improved in-context learning. We introduce a novel two-stage approach that optimizes examples for improved in-context learning. To the best of our knowledge, this is the first work to do this. As Reviewer VRBk puts it, ''the overall goal of using VLMs to distill demonstrations into canonical examples is an interesting combination of foundation models with classical ideas such as case-based learning.'' Our results, in Tables 1-3, demonstrate that both stages are crucial for high performance, significantly outperforming CoT and RAG, which use raw or handwritten in-context examples. Additionally, Figure 5 illustrates our method's support for continual learning without forgetting.

Furthermore, our optimized in-context examples are applicable to various multimodal agents across three complex domains: long-horizon instruction-following for household robots, visually-grounded web navigation, and action forecasting from real-world egocentric videos. Reviewers vg3K and VRBk note that these experiments on "sufficiently different tasks... suggest the generalizability of the proposed method." Previous works have been limited to text-based environments, gaming scenarios, or single real-world domains. To our knowledge, we are the first to show robust results across these three diverse and complex domains.

---

### Decision · Program_Chairs · 2024-09-25

**Decision:**

Accept (spotlight)

**Comment:**

This work proposes a technique that leverages large-scale models to generate experience abstractions (or canonical examples) from sub-optimal demos and feedback from humans, bringing ideas of case-based learning and foundational models together. This, in turn, helps adapt the agent to perform better across a range of benchmarks/tasks -- Teach, Visual Web Arena, and Ego4D.

The committee highlighted several strengths, particularly the novelty of the method. The committee found the paper effectively demonstrates broad utility, as empirically shown across three domains of tasks. This work has a high impact as an alternative pathway to the reliance on labels/demos from experts.

The discussion phase led to further clarifications that helped solidify the committee's positive outlook on the work. The AC concurs with the reviewers' positive assessment and recommends acceptance. The authors are encouraged to utilize their constructive and detailed discussion with reviewers to further raise the quality of this submission.